# Permafrost in the Cretaceous supergreenhouse

**Juan Pedro Rodríguez-López**[1,2,3]**, Chihua Wu**[1,4] ✉**, Tatiana A. Vishnivetskaya** [5], **Julian B. Murton** [3]**, Wenqiang Tang**[1,6] **& Chao Ma**[4]

Earth's climate during the last 4.6 billion years has changed repeatedly between cold (icehouse) and warm (greenhouse) conditions. The hottest conditions (supergreenhouse) are widely assumed to have lacked an active cryosphere. Here we show that during the archetypal supergreenhouse Cretaceous Earth, an active cryosphere with permafrost existed in Chinese plateau deserts (astrochonological age ca. 132.49–132.17 Ma), and that a modern analogue for these plateau cryospheric conditions is the aeolian–permafrost system we report from the Qiongkuai Lebashi Lake area, Xinjiang Uygur Autonomous Region, China. Significantly, Cretaceous plateau permafrost was coeval with largely marine cryospheric indicators in the Arctic and Australia, indicating a strong coupling of the ocean–atmosphere system. The Cretaceous permafrost contained a rich microbiome at subtropical palaeolatitude and 3–4 km palaeoaltitude, analogous to recent permafrost in the western Himalayas. A mindset of persistent ice-free greenhouse conditions during the Cretaceous has stifled consideration of permafrost thaw as a contributor of C and nutrients to the palaeo-oceans and palaeo-atmosphere.

Permafrost is an amplifier of climate change[1,2], emitting $CO_2$ and $CH_4$ from bacterial carbon degradation as permafrost thaws[3], and providing nutrients and carbon to aquatic ecosystems[4,5]. But evidence of permafrost in the pre-Quaternary geological record ('deep time') is limited, as is geological evidence of continental ice during supergreenhouse periods. Intriguingly, however, a growing body of evidence suggests that cryospheric conditions developed at different times during the Cretaceous supergreenhouse.

Most evidence of a Cretaceous cryosphere comes in the form of marine ice-rafted debris (IRD) from the southern[6,7] and northern hemispheres[8–11], as well as from the recognition of marine glendonites (calcite pseudomorphs after ikaite) indicating cold temperatures during formation[12–15]. Additional evidence includes Cretaceous landforms in Yukon and Alaska[16], patterned ground in China[17], and ice-rafted dropstones in desert oases indicating cold-desert conditions in interdunes similar to those in the Badain Jaran Desert[11]. Recently, the discovery of ultra-depleted hydrogen isotopes from Antarctica has suggested glaciation of the South Pole during the Late Cretaceous[18], and there is isotopic evidence for continental ice sheets in China during the Early Cretaceous[19]. Collectively, these studies hint at the likely occurrence of Cretaceous periglacial and permafrost environments in polar and high-altitude regions.

Here we report the former occurrence of Cretaceous permafrost in a plateau desert in China analogous to modern permafrost in the Western Himalayas (Fig. 1a). This raises new questions on the relative role of allogenic controls on Mesozoic palaeoclimates, including a cryosphere and the widespread release of nutrients to the palaeo-

[1]PAGODA Research Group (Plateau & Global Desert Basins Research Group), Institute of Sedimentary Geology, Chengdu University of Technology, 610059 Chengdu, China. [2]Department of Geology, Faculty of Science and Technology, University of the Basque Country (UPV/EHU), Ap. 644, E-48080 Bilbao, Spain. [3]Permafrost Laboratory, Department of Geography, University of Sussex, Brighton BN1 9QJ, UK. [4]State Key Laboratory of Oil and Gas Reservoir Geology and Exploitation, Institute of Sedimentary Geology, Chengdu University of Technology, 610059 Chengdu, China. [5]Center for Environmental Biotechnology, University of Tennessee, Knoxville, TN 37996, USA. [6]Research Institute of Petroleum Exploration and Development, PetroChina Southwest Oil and Gas Field Company, 610051 Chengdu, Sichuan, China. ✉e-mail: wuchi-hua@foxmail.com

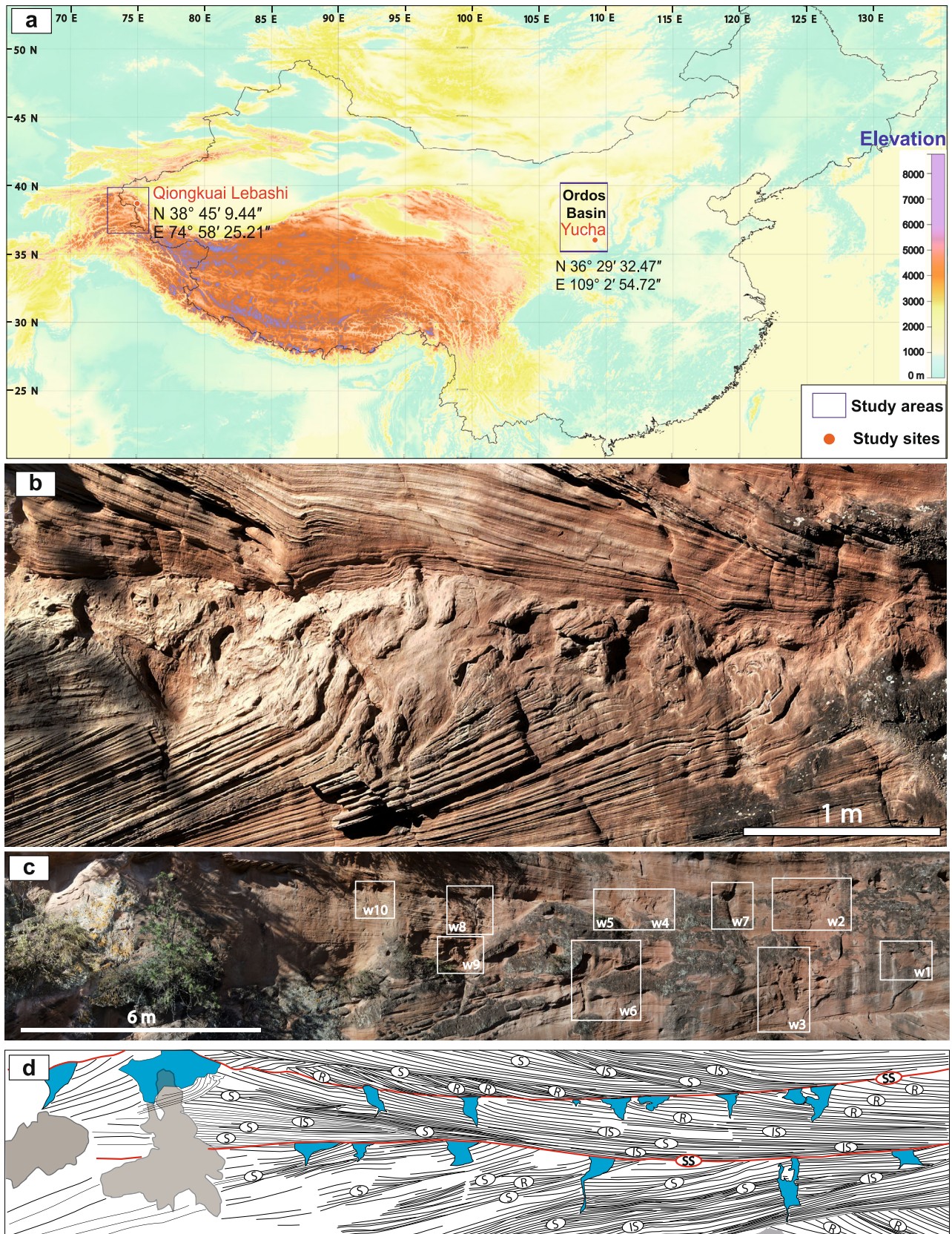

Arctic realm. The identification of past permafrost is crucial for understanding climate dynamics and the ocean–continent coupling during supergreenhouse conditions. Of particular importance is the response of high-altitude permafrost thaw due to global warming.

Permafrost thaw in the eastern Tibetan Plateau between 1969 and 2017, has increased by a factor of 40, with 70% of the thawed area forming since 2004, amplifying large-scale permafrost climate feedbacks[20].

**Fig. 1 | Cretaceous permafrost wedges. a** Elevation map of the Himalayas and the Tibetan Plateau showing the location of the study sites in the Ordos Basin (Cretaceous permafrost) and the Qiongkuai Lebashi Lake (recent permafrost analogue). The Digital Elevation Model (DEM) data were downloaded from the Shuttle Radar Topography Mission (SRTM) on the USGS EarthExplorer (https://earthexplorer.usgs.gov/). Image was generated by using the Global Mapper 18.0 (Blue Marble Geographics) programme based on the Digital Elevation Model data. **b** Permafrost sandstone wedges horizon covered and preserved by downlapping aeolian dune toeset sandstones. **c** Field photograph of two wedge horizons hosted in aeolian dune sandstones of the Luohe Fm. **d** Ten wedges are identified (labelled wedge 1 "w1" to wedge 10 "w10" in **c**). Detailed sedimentological observations of the permafrost wedges can be seen in Fig. 2 and Supplementary Fig. 6. Aeolian architecture in **d** is based on the recognition of aeolian bounding surfaces hierarchy[25]; "SS" aeolian supersurface; "IS" interdune surface; "S" superimposition surface; "R" reactivation surface. Wedges are marked in blue colour. See the enlarged image in Supplementary Fig. 5.

## Results

### Cretaceous permafrost wedges and host desert dunes

Early Cretaceous westerly winds blew in eastern China[21] forming gigantic aeolian dunes (>352 m high)[22] in the sandy deserts (ergs) represented by the 110–430 m-thick Luohe Formation (Fm) in the Ordos Basin[22] (Supplementary Notes 1 and 2, and Supplementary Figs. 1a, b and 2a). This formation comprises aeolian cross-bedded sandstones (Fig. 1b, and Supplementary Figs. 3–5) preserved in the extensional Ordos Basin (North China Craton)[23] that formed after the late Mesozoic subduction of the Palaeo-Pacific Plate and the closure of the Mongolia–Okhotsk Ocean[24] (Supplementary Notes 1 and 2). The studied outcrops of the Luohe Fm occur near Yayodi Village, Yucha Grand Canyon, Shaanxi Province, China (Fig. 1a, and Supplementary Fig. 1b–d). The preservation of the Cretaceous desert aeolian dune fields (ergs) is exceptional. They exhibit a stratigraphic architecture commonly observed in other recent and fossil *draas* (complex dunes)[25] (Supplementary Note 2) characterized by a well-defined hierarchy of aeolian bounding surfaces, including reactivation, superimposition, interdune (interdraa) and supersurfaces (Fig. 1b–d, and Supplementary Figs. 3–5). The supersurfaces are erg sequence boundaries[25] and are commonly associated with horizons of sandstone wedges, indicating a common genetic origin (Fig. 1b–d and Supplementary Fig. 5a–c).

Sandstone wedges have been identified in three different outcrops of the Luohe Fm. Outcrop one (Fig. 1b, and Supplementary Figs. 4 and 6e) shows two distinct levels of wedges separated by trough-cross bedded aeolian sandstones with tangential downlapping of aeolian toeset sediments on the wedge tops. Outcrop two (Supplementary Figs. 3 and 6a–d) shows two wedges penetrating aeolian dune cross-bedded sets. Outcrop three (Figs. 1c, d, 2a–e and Supplementary Fig. 5) shows 10 wedges concentrated in two discrete horizons bounding three draa successions (Fig. 1c, d).

The sandstone wedges from the Cretaceous Luohe Fm show evidence of both primary and secondary infilling during the formation of thermal contraction crack wedges in permafrost and periglacial environments (Tables 2 and 3 of refs. 26,27 (Supplementary Note 3)). Primary infilling of open thermal contraction cracks by aeolian sand is indicated by vertical to subvertical lamination within many of the sandstone wedges. Individual sandstone veins that branch away from the toes of the wedges represent individual crack infills. The same sedimentary structures typify cryogenic veins and wedges of primary infilling in present-day Arctic and Antarctic regions[28], mid-latitude Pleistocene permafrost regions[29,30], and in Proterozoic sandstones in Australia[31,32].

Secondary infilling with sand of voids created by the melt of original ice veins or small ice wedges within the overall wedge forms is indicated by several lines of evidence. First, small normal faults, some step-like, in strata adjacent to some wedges, indicate extensional faulting during the melt of ice[33]. Second, collapse structures[34], and involutions in the upper part of some wedge infills indicate subsidence of adjacent sediment from the sides or roof during the melt of ice veins[35,36]. Third, fallen and rotated intraclasts were derived from the host sediments once adjacent to the upper margins of the original wedge[33]. Fourth, some parts of the infills appear massive, either because secondary infilling has disrupted the original primary

lamination or because the lamination did not develop due to very uniform particle size (Fig. 2, and Supplementary Fig. 6).

Some sandstone wedges show a polygenic infill characterized by multiple wedges crossing horizontal lamination preserved in the wedges (Fig. 2). These complex patterns and superimposed wedges are similar to those in Upper Pleistocene wedges in western Europe (ref. 33, and references therein) and northwest Russia[37]. Superimposition results from the reactivation of thermal contraction cracking when the appropriate thermal conditions resume after a period of inactivity[38].

Collectively, the sedimentary properties of the sandstone wedges in the Luohe Fm indicate that many of the wedges represent composite-wedge pseudomorphs, because they have composite infillings that comprise evidence for both primary and secondary infilling[26,39] (Figs. 1b, 2, and Supplementary Fig. 6). Wedges lacking evidence for secondary infilling are interpreted as relict sand wedges. Although small sand veins and wedges up to 0.5 m wide and up to 1.2 m deep may develop in regions of permafrost and deep seasonal frost[40], the presence of composite-wedge pseudomorphs strongly supports the interpretation that they developed in a permafrost environment. This interpretation is also consistent with the dimensions of the sandstone wedges (up to at least 2 m high, and up to ~1 m wide), dimensions that are common in modern permafrost environments[28] but have not been demonstrated to form purely under conditions of seasonal freezing. Overall, therefore, it is highly likely that the sandstone wedges of the Luohe Fm developed under conditions of past permafrost (Supplementary Note 3).

A late Pleistocene analogue for the Cretaceous aeolian–permafrost system of the Luohe Fm is provided by the composite wedges and sand wedges within aeolian dune deposits of the Kittigazuit Fm., Hadwen Island, NT, Canada[41]. Analysis (Supplementary Note 4) of the sedimentological and architectural analogies demonstrates multiple features shared by both the aeolian–permafrost systems from the Canadian Pleistocene and the Chinese Cretaceous (Supplementary Figs. 7–9).

### Recent permafrost analogue from the Western Himalayas

The evidence of past permafrost in a desert basin during the Cretaceous supergreenhouse period poses an apparent conundrum, but one that can be explained by reference to a modern analogue of a high-altitude aeolian–permafrost system at Qiongkuai Lebashi Lake, Xinjiang Uygur Autonomous Region, in the western Himalayas, China (Figs. 1a, 3–6, Supplementary Note 5 and Supplementary Fig. 10). Here, evidence from satellite imagery of permafrost persisting in the plateau lake–aeolian system for a period of 8 years (Fig. 3c–f, and Supplementary Fig. 10) corroborates the permafrost modelling for the study area[42,43] (Fig. 5), and the measured altitudes for the aeolian dunefield and associated frozen oases of 3308 m above sea level (asl) (Figs. 4, and 6a–d) agrees with the probability of permafrost occurrence at this altitude[44] (Fig. 6a–d). Satellite imagery and topographic elevation sections of the study area (Fig. 6a) show that the terrain near Qiongkuai Lebashi Lake varies in altitude from 3.3 to 5.1 km (asl) (Fig. 6a).

The development of Cretaceous permafrost composite wedges and sand wedges associated with major bounding surfaces

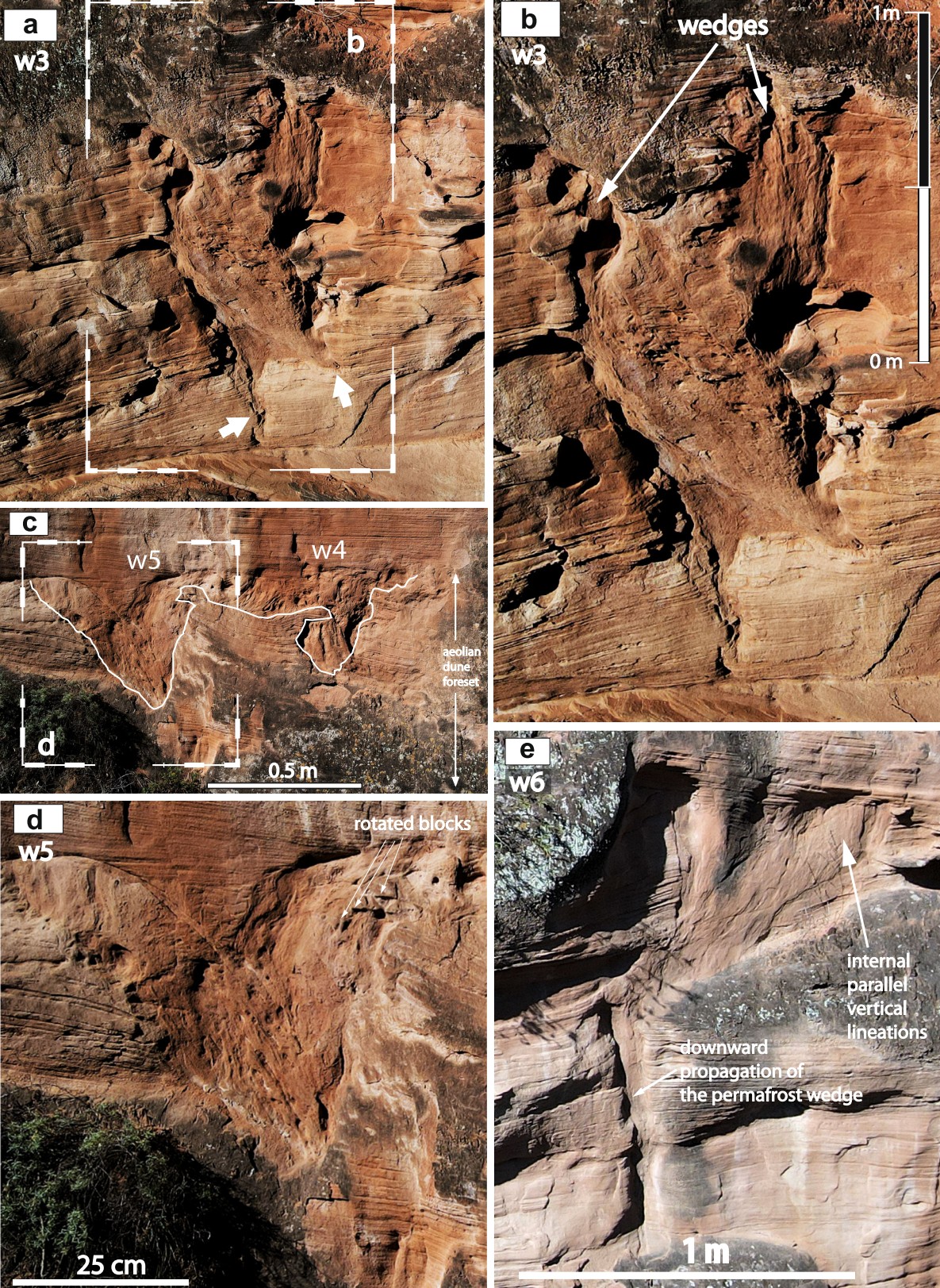

**Fig. 2 | Sedimentology of sandstone wedges in the aeolian dunes of the Luohe Fm.** See location of wedges in Fig. 1c and Supplementary Fig. 5b. **a** Wedge w3, white arrows mark the termination of the wedge toes in the cross-bedded aeolian dune sandstones. **b** Close-up view from **a** showing two wedges and their downward terminations. **c** Wedges w5 and w4, and **d** close-up view from **c**, showing rotated blocks of host sandstones into the margin of the wedge and overlying grain-flow facies downlapping and burying the wedge. **e** Wedge w6 showing internal parallel vertical lamination and downward propagation of wedge toe into host aeolian sandstones. The wedge is sharply overlain by laminated aeolian sandstones.

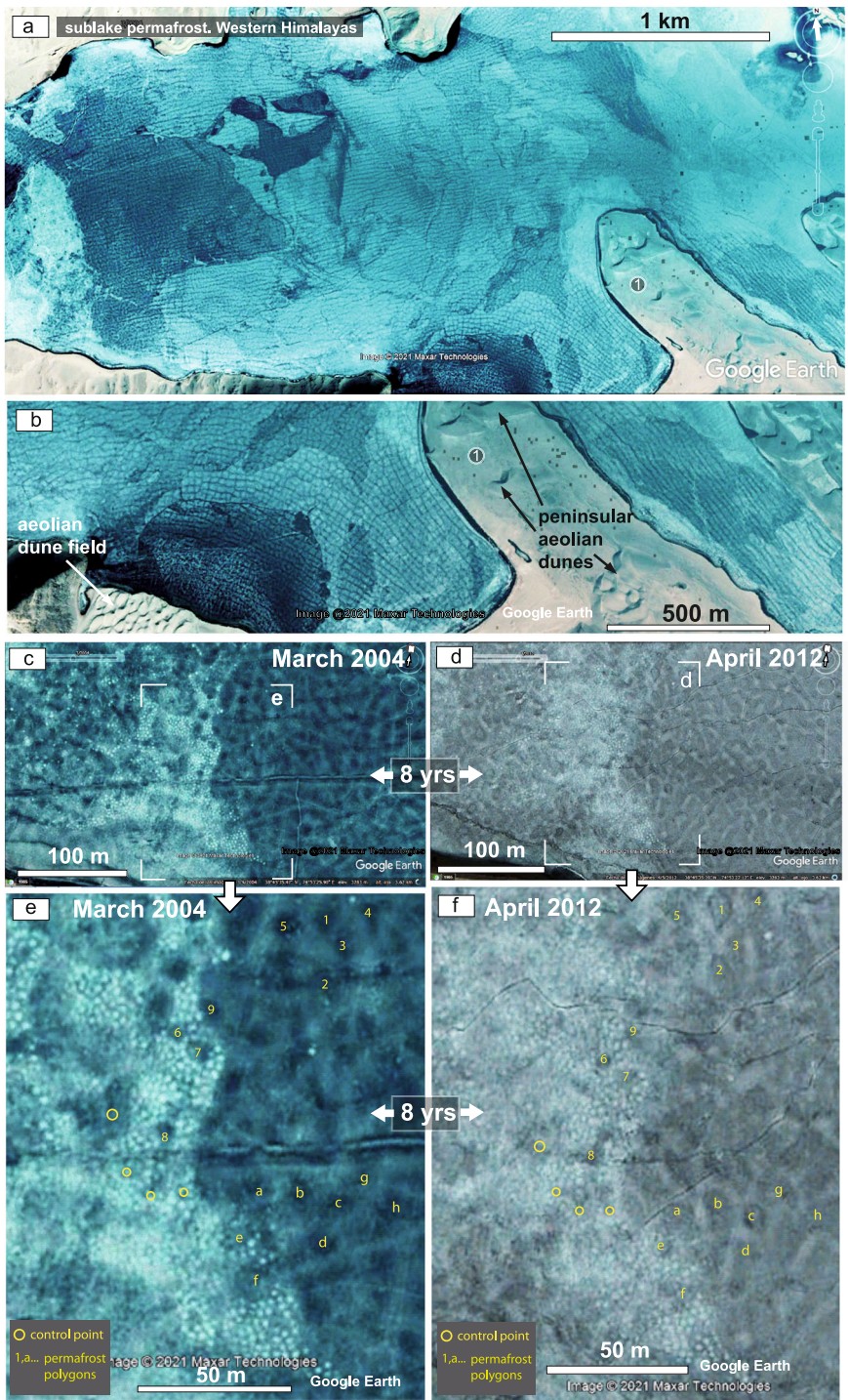

**Fig. 3 | Persistent permafrost in the Qiongkuai Lebashi Lake, Xinjiang Uygur Autonomous Region, China. a** Evidence of widespread sublake permafrost polygon network visible through clear lake ice. **b** Note how fractures within lake ice do not affect the underlying permafrost polygon pattern. **c–f** Evidence of sublake permafrost patterned ground persisting for 8 years in the lake. **c**, Satellite imagery from March 2004, and **d** satellite imagery from April 2012 showing the same area of the lake. **e** and **f** show the mapping of the individually identified permafrost polygons persisting over a time period of 8 years. Rounded isomorphic white dots in the satellite image are interpreted as permafrost features known as earth hummocks. See text for discussion (**a–f** from Image@2021 Maxar Techologies and Google Earth).

(supersurfaces) in the Ordos Basin (Fig. 1c, d) is compatible with variations on the desert-basin phreatic level as those recorded in satellite imagery of Qiongkuai Lebashi Lake (Supplementary Note 5). There, both permafrost persisting for at least 8 years (Fig. 3c–f, and Supplementary Fig. 10), and coeval lake transgression led to the recognition of permafrost affecting the interdunes and margins of the aeolian dunefields (Fig. 4) at an altitude of 3–4 km asl.

Permafrost processes are active on plateau lacustrine–aeolian dunefields near Qiongkuai Lebashi Lake (Figs. 4–6). Sublake permafrost polygons are visible in the shallow bottom of the lake (Fig. 3a and b), and despite the transgression observed in the lake in 2007 (Supplementary Fig. 10), the sublake permafrost polygons persisted for a period of 8 years (2004–2012) (Fig. 3c–f, and Supplementary Fig. 10). Satellite imagery reveals patterned ground, showing some regular

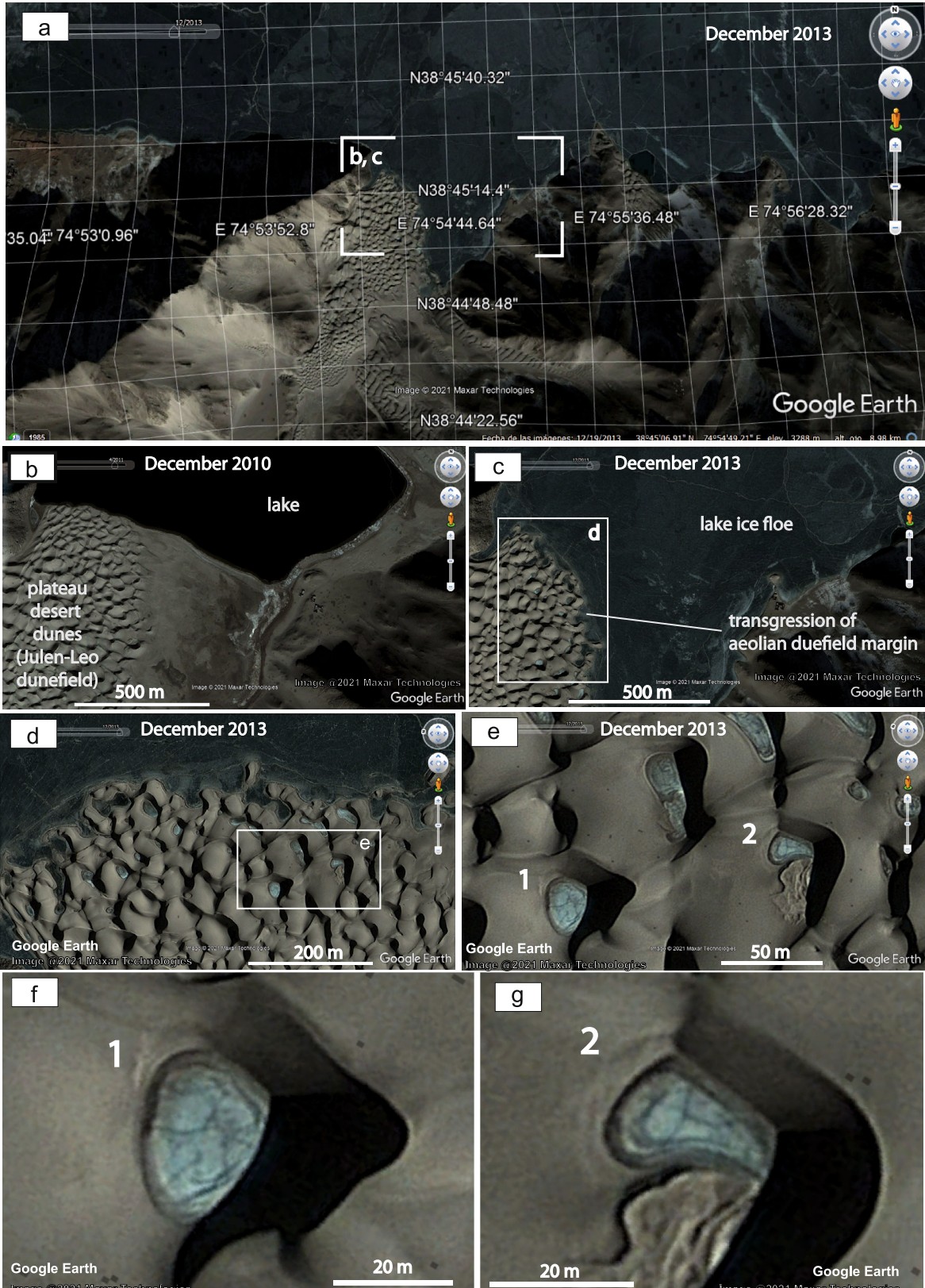

**Fig. 4 | Close-up views of the plateau cold aeolian–lake system in the western Himalayas. a** Southern margin of the Qiongkuai Lebashi Lake. **b** Image dated December 2010, and **c** Image dated December 2013, showing evident transgression of the lake shore and onto the aeolian dunefield. **d** Close-up view from **c**, rotated indicating the geographic north to the left of the image. See location in (**c**). **e** Close-up view from **d** showing frozen interdunes and possible patterned ground. **f** and **g** Close-up view of the frozen interdunes showing patterned ground. See 1, and 2 labelled frozen oases in (**e**) (**a**–**g**: Image @2021 Maxar Technologies and Google Earth).

orthogonal (rectangular) thermal contraction crack polygons adjacent to the western and southern shorelines (Fig. 3). The lake ice looks to be melting, with ice-free (very dark) areas at the lake margin (Fig. 3a and b). The high transparency of the lake ice with no snow cover allowed good visibility of lake bottom permafrost polygons (Fig. 3).

In addition to the large-scale polygonal patterns, small-scale patterns are visible in the form of persistent rounded structures (Fig. 3e and f). These structures have dimensions and shapes similar to those of earth hummocks, which are commonly observed in arctic permafrost terrain[45,46]. The same hummocks are visible in 2004 and 2012 associated with the permafrost polygons (Fig. 3c–f), providing further evidence of permafrost persisting beneath the bottom of the shallow plateau lake. Such persistence of hummocks and polygons indicates beyond reasonable doubt that these parts of the lake were sufficiently shallow to freeze to the bottom in winter, enabling permafrost to persist beneath a lake-bottom active layer (Fig. 3). As a result, the patterned ground was preserved.

Lastly, frozen oases (interdunes) also show polygonal patterns (Fig. 4a–g). Considering that both the shallow lake water and the interdune waters are frozen, and permafrost patterned ground is preserved beneath parts of the lake, then the water table in the aeolian dunefield must be also frozen, defining permafrost in the subsurface of the aeolian dunefield.

The occurrence of discrete horizons showing permafrost sandstone wedges (Figs. 1b–d, 2, and Supplementary Fig. 5) in a Cretaceous plateau aeolian dunefield is consistent with a long-lasting permafrost plateau aeolian system similar to that observed recently in the Qiongkuai Lebashi Lake area, Xinjiang Uygur Autonomous Region, China, where during transgressive periods the plateau lake favoured permafrost persisting in aeolian interdunes (Fig. 4d–g). Well-defined permafrost horizons recorded in the lower Cretaceous Luohe Fm (Fig. 1c and d, and Supplementary Fig. 5) attest to a similar process, where persistent plateau freezing conditions allowed the development of composite wedges and sand wedges in an aeolian dunefield. The occurrence of these permafrost wedge horizons associated with aeolian supersurfaces is further evidence of the temporal relationship among permafrost development and a variable phreatic level controlled by allogenic forcing[47], as demonstrated in the satellite images of the recent analogue from the western Himalayas (Fig. 4b–g).

The satellite imagery evidence of modern permafrost at Qiongkuai Lebashi Lake is consistent with a map that combines a digital elevation model (DEM) with modelled permafrost probability (PERPROB) data from refs. 42,43. The map indicates that this area is probably underlain by permafrost; PERPROB values of 0.5–0.9 are classified as discontinuous permafrost, and values of >0.9 as continuous permafrost[43] (Figs. 5a–d, and 6c).

Regionally, the statistical distribution of modern high-altitude permafrost, glaciers, and the zero-degree quantiles for high-mountain Asia (Fig. 6b)[44] indicates that permafrost presently underlies 2–3.7% of the terrain at altitudes matching those of the cold aeolian dunefields and surrounding relief (3.3–3.6 km asl) and 7.3–11.8% of the surrounding glaciated mountain ranges (4.5–5.2 km asl) (Fig. 6a, b). The occurrence of Cretaceous permafrost in the plateau aeolian system supports the palaeotemperature modelling for this area for this time (ca. 0 °C)[48] (Supplementary Fig. 2b), and two scenarios are proposed for the Cretaceous palaeolatitude of the plateau desert system in the Ordos Basin: 53°N for the Barresian–Valanginian[48], and 32.6–41.0°N for the Early Cretaceous[49,50] (Fig. 6e). A palaeolatitude of 53°N for the Cretaceous permafrost is similar to the latitudinal distribution of modern permafrost and glaciers in the North Asia area defined by the IPCC, and a palaeolatitude of 41°N for it is similar to the latitudinal distribution of permafrost and glaciers in the High Mountain Asia area defined by the IPCC (Fig. 6e). The 53°N scenario suggests an elevation mode of past permafrost at 899 m asl (range 186–3841 m asl), with permafrost underlying 17% of the surface (Fig. 6e). The 41°N scenario yields an elevation range for permafrost of 2196–6204 m asl, with a mode of 4672 m asl, and permafrost underlying 12% of the surface (Fig. 6e).

## Permafrost astrochronology and the orbital control on the Hauterivian cold snap

The revised magnetostratigraphic studies were based on the geomagnetic polarity time scale (MHTC12)[51] and cyclostratigraphic analysis (Supplementary Note 6). The observed magnetic polarities are correlated with chrons CM5n–CM12r.1n of the geomagnetic polarity time scale MHTC12, yielding the age range of 134.0–126.1 Ma for the measured section (Supplementary Note 6 and Supplementary Fig. 11). The top boundary of the Luohe Fm is ca. 129.4 Ma. The Luohe Fm of Well Wuqi created a ca. 4.19 myr long floating astronomical time scale (sedimentation rate: 8.243 cm/kyr, dominated cycle: long eccentricity with the frequency of 33.09 m/cycle; Supplementary Note 6 and Supplementary Fig. 12), and the astronomical age of the Well Wuqi is ca. 133.59–129.4 Ma during the Luohe Fm interval (Supplementary Note 6).

Some notable low-power eccentricity (E7 and E8) occurs in the 405-kyr Gaussian bandpass filter (Fig. 7). The eccentricity signal during this period also became weaker in the evolutionary fast Fourier transform (FFT) spectral analysis of Well Wuqi (Fig. 7). In addition, an extended period of low-amplitude variability in obliquity is identified in the lower part of the Luohe Fm (ca. 132.49–132.17 Ma) (Fig. 7).

A similar phenomenon has been documented for the Mi-1 glaciation event[52]. The lower the obliquity, the less solar radiation the polar regions receive, which favours the creation of ice sheets[52,53]. Compared to Mi-1, the extent and duration of cooling reflected in our records are weaker and shorter. Thus, we presume that from ca. 132.49 Ma onwards, the polar summer continued to be cooler, and the ice sheets temporarily expanded with the appearance of minimum eccentricity and a sustained low amplitude in obliquity[52,53]. The evolutionary FFT spectrum and the power of eccentricity and obliquity can be correlated with the horizon of the permafrost wedges, which supports the accuracy of the revised palaeo-magnetic age framework (Fig. 7).

## Cretaceous permafrost microbiome

A diverse complex of recognizable, and often exceptionally well-preserved ancient filamentous, bacillar, and coccoid fossilized microorganisms was found in the Cretaceous permafrost wedges (Fig. 8 and Supplementary Figs. 13–16). Microbial fossils in the sample OR37b4 appeared to be more diverse in comparison to sample OR15b1 (Fig. 8). The fossils include remnants of prokaryotes (rod-shaped and coccoid bacteria, hyphae of actinomycetes, filamentous cyanobacteria), and aquatic eukaryotic unicellular microalgae (e.g., prasinophytes). The permafrost biome shows mineralized eukaryotic microorganisms gathered in monolayers where microbial cells were enveloped in sheaths and formed a film containing numerous oval cells about 1.4 μm long and 1–1.2 μm wide. Rounded cells with a diameter of about 1.4 μm and oval-flattened cells are also visible (Fig. 8a–c). Rod-shaped bacterial forms are 2.5–3.5 μm long and 0.4–0.5 μm wide, slightly curved with rounded ends and resemble Bacilli (Fig. 8d–f). All these sizes are in agreement with the most common dimensions of fossil bacteria, within the range of 0.2–2 μm, although maximum sizes of fossil bacteria can be >100 μm[54].

Several lines of evidence indicate the mineralized stage of fossil bacteria and its fossilization. First, a lack of morphological variation and neither dehydration nor shrinkage of cells indicates a mineralized stage of fossil bacteria, as demonstrated by SEM images taken before and after the EDX analysis (Supplementary Figs. 13 and 15). Second, the geochemical composition of analysed fossil bacteria reveals a C and O composition (carbon ca. 16–44%, oxygen ca. 25–43%) different from that of living organisms but very similar to lignite coal, peat and humic substances (Supplementary Figs. 14 and 16). Third, geochemical EDX analysis demonstrates that this high C content occurs in fossils already

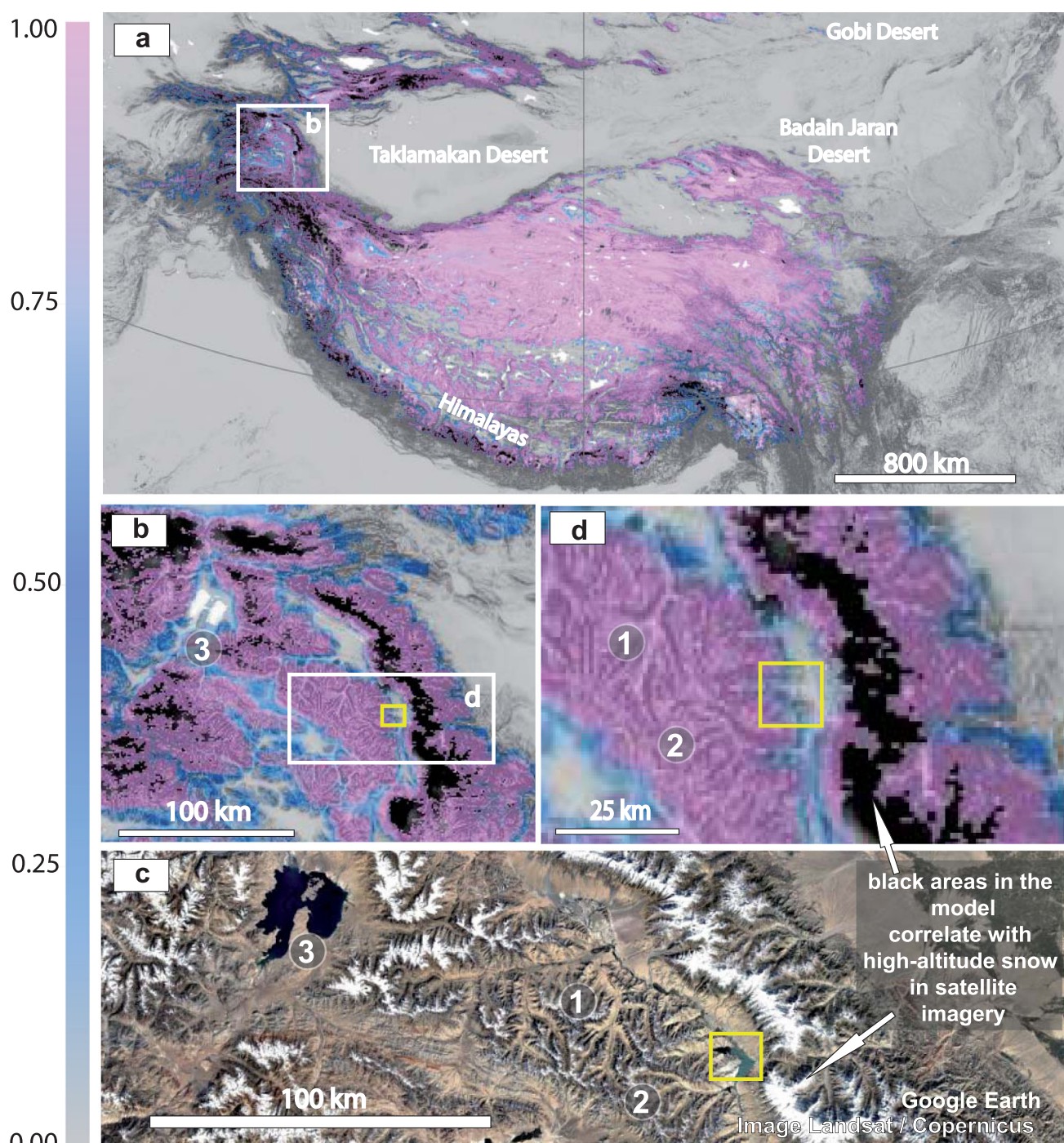

**Fig. 5 | Permafrost probability fraction (PERPROB) maps of modern permafrost distribution in the Himalayan–Tibet Plateau region. a, b**, and **d** Maps producing by combining a DEM with Permafrost Probability Fraction (PERPROB) data from refs. 42,43 https://doi.pangaea.de/10.1594/PANGAEA.888600?format=html# download, using GlobalMapper software. The colour ramp showing PERPROB values is on the left of the figure. **c** Satellite imagery indicating the location of the study area in the western Himalayas. Image Landsat/Copernicus and Google Earth.

mineralized with high-weight elemental concentrations of O, Si and Ca (Supplementary Figs. 14 and 16), demonstrating the mineralized stage (calcitization and silicification) of fossil bacteria[55]. Calcitization and silicification are two early diagenetic/taphonomic processes leading to the exceptional preservation of fossil bacteria[56]. Silicification leads to the excellent preservation of cells[56], as shown by the preservation of the Cretaceous permafrost biome cells with high-weight elemental concentrations of O and Si, similar to the exceptionally preserved bacteria in Proterozoic rocks from the Mesoproterozoic Wumishan Fm in Jixian, north China[55].

Evidence of synsedimentary growth of bacterial colonies in the aeolian facies that host the sandstone wedges during deposition includes the widespread occurrence of mineral grains and cements covering microbial fossilized cells[54] (Fig. 8). Based on previous research[57,58] we summarize that gram-positive bacteria, cyanobacteria, and unicellular eukaryotic algae are more often found as fossilized remains. Bacterial colonies, as those from the Cretaceous permafrost in China, indicate that the permafrost microbiome was organized in communities for efficient exploitation of energy and nutrients[54] in this extremely cold environment, where permafrost thaw might hamper

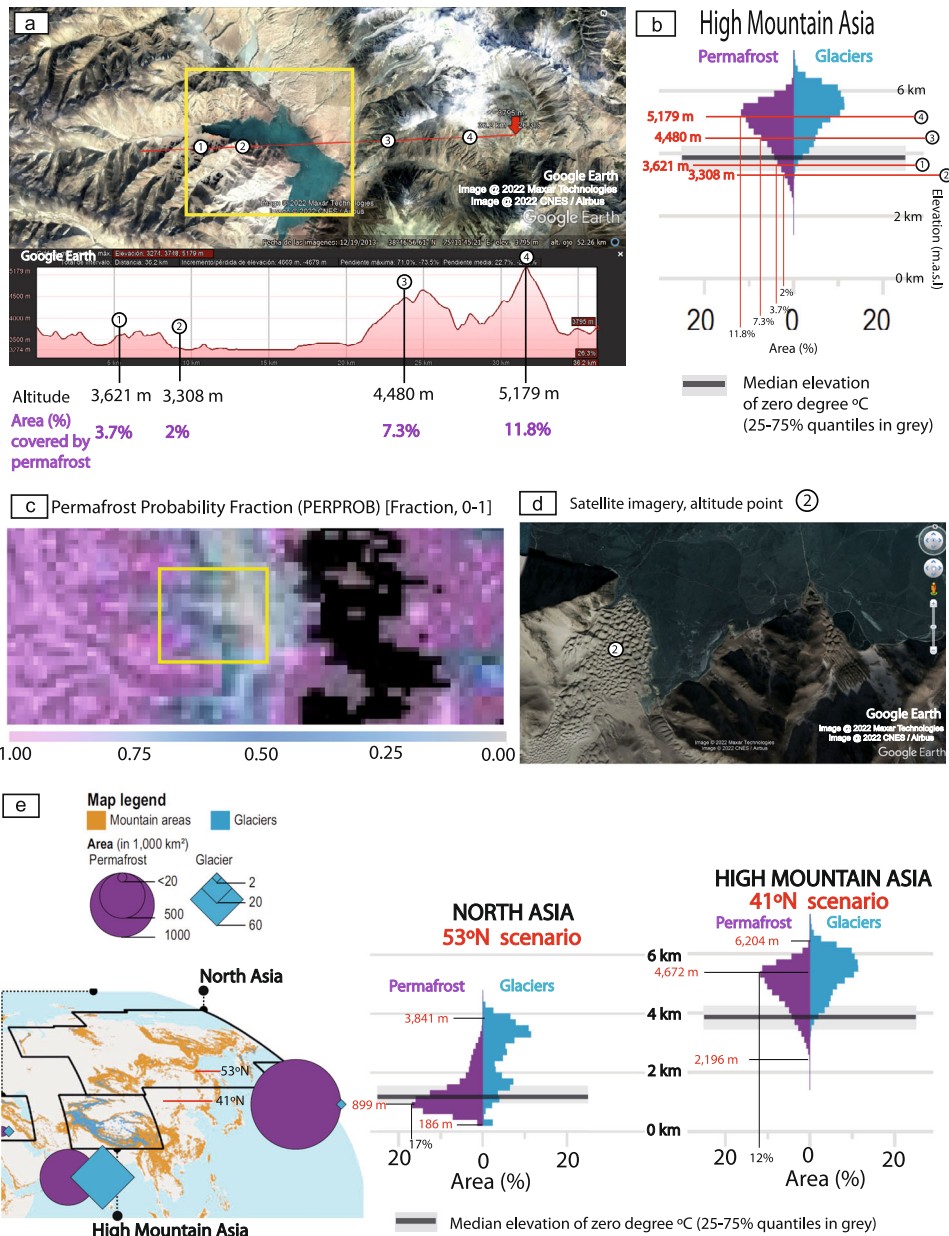

**Fig. 6 | Altitudinal and geographical distribution of high-altitude Himalayan permafrost. a** Satellite imagery and topographic elevation profile of the cold aeolian dunefield, high mountain lake and surrounding mountain systems. Image @2022 Maxar Technologies, Image @ CNES/Airbus, and Google Earth. **b** Regional summary statistics for the area of glaciers and permafrost in mountains of the High Mountain Asia region and the median elevation of the annual mean 0 °C free-atmosphere isotherm with 25–75% quantiles in grey. Adapted from Fig. 2.1 from ref. 44. Adapted from Fig. 2.1 from Hock, R., G. Rasul, C. Adler, B. Cáceres, S. Gruber, Y. Hirabayashi, M. Jackson, A. Kääb, S. Kang, S. Kutuzov, Al. Milner, U. Molau, S. Morin, B. Orlove, and H. Steltzer, 2019: High Mountain Areas. In: IPCC Special Report on the Ocean and Cryosphere in a Changing Climate [H.-O. Pörtner, D.C. Roberts, V. Masson-Delmotte, P. Zhai, M. Tignor, E. Poloczanska, K. Mintenbeck, A. Alegría, M. Nicolai, A. Okem, J. Petzold, B. Rama, N.M. Weyer (eds.)]. In press. The altitudes of

selected geomorphological elements (1–4) from **a** are indicated, as well as their correspondent % of the area covered by permafrost. **c** Maps produced by combining a DEM with permafrost probability fraction (PERPROB) data from refs. 42,43 https://doi.pangaea.de/10.1594/PANGAEA.888600?format=html#download, using GlobalMapper software. **d** Satellite imagery showing the location measured point, 2 in the plateau cold aeolian dunefield. See location in (**a**). Image @2022 Maxar Technologies, Image @ CNES / Airbus, and Google Earth. **e** Comparison of the palaeolatitudinal reconstructions of the studied Cretaceous permafrost system from China (53°N and 41°N scenarios), with the statistics on the present-day altitudinal distribution of permafrost, glaciers, and zero-degree quantiles in North Asia and High Mountain Asia from Fig. 2.1 of ref. 44. Adapted from Fig. 2.1 from Hock, R. et al., 2019: High Mountain Areas. In: IPCC Special Report on the Ocean and Cryosphere in a Changing Climate (eds Pörtner, H.-O. et al.) (in press).

trophic or cooperative interactions in the long-term among different microorganisms colonizing the permafrost[59].

The fossils found in the Cretaceous permafrost wedges include prokaryotes (rod-shaped and coccoid bacteria, hyphae of actinomycetes, filamentous cyanobacteria), and aquatic eukaryotic unicellular microalgae (e.g., prasinophytes), resembling microbiomes from late Pliocene through Holocene permafrost[60,61]. Analyses of

shotgun metagenomes from modern permafrost and the overlying active layer in Alaska[62], in the High Arctic (Svalbard)[63], and Canada[64] showed the presence of actinomycetes, filamentous cyanobacteria and eukaryotic algae. Siberian permafrost contains cyanobacteria and unicellular eukaryotic algae that can survive harsh permafrost conditions for millennia[61]. The preserved Cretaceous permafrost microbiome assemblage contains decomposers of organic matter

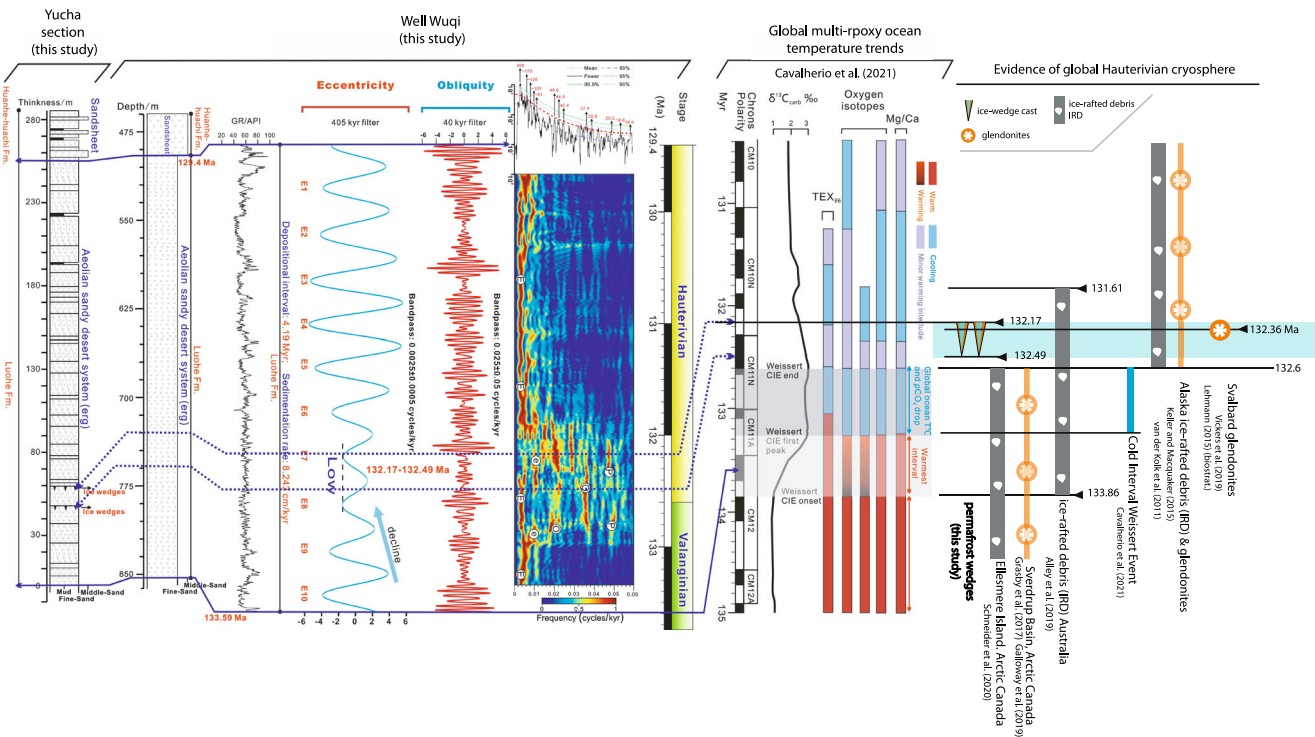

**Fig. 7 | Chronostratigraphic interval containing permafrost sandstone wedges based on the astrochronology of Well Wuqi in the Ordos Basin, and correlation with global multi-proxy ocean temperature trends and with other evidence of and active Cretaceous cryosphere.** The $\delta^{13}Ccarb$ curve, TEX$_{86}$, oxygen isotopes, and Mg/Ca ratio palaeo-thermometry from ref. 68 and references therein. Modified after ref. 68. Temporal distribution of Svalbard glendonites from ref. 65 and biostratigraphy based on ref. 66 after chronostratigraphy of ref. 65. Alaskan IRD and glendonites from ref. 67,75 and Australian IRD from ref. 6. Glendonites from the Deer Bay Fm in the Sverdrup Basin, Arctic Canada from refs. 12,13. Dropstones from the Deer Bay Fm in Ellesmere Island, Arctic Canada from ref. 9.

such as actinomycetes, mycelial fungi, and bacteria, along with photosynthetic cyanobacteria and unicellular algae that controlled the net carbon dioxide emissions during Cretaceous permafrost thaw.

## Discussion

Permafrost underlay a plateau desert in interior Asia during the Cretaceous supergreenhouse and was analogous to modern permafrost in a desert in the western Himalayas surrounding Qiongkuai Lebashi Lake. Identification of this past permafrost corroborates the palaeo-temperature models for the Valanginian–Hauterivian stages[48] that predicted an annual mean surface air temperature (at 1.5 m) of ≤0 °C for the Ordos Basin plateau desert basin and −8 °C for the bounding desert mountain ranges (Supplementary Fig. 2b). The age of the permafrost sandstone wedges from the Luohe Fm is ca. 132.49–132.17 Ma (Hauterivian), based on the astronomical age constraint, and correlates with the occurrence of glendonites in Svalbard[65], ca. 132.36 Ma [boundary between *B. Balearis* and *P. Ligatus* biozones[66]], ice-rafted debris (IRD) in Australia[6], and glendonites and IRD in Alaska[67], and slightly later than the Weissert event[68] (Fig. 7). The occurrence of well-known Valanginian-age glendonites[13], and IRD[9], together with the identification of geochemical anomalies in western Tethys[69] points to extensive glacial ice in the Valanginian–Hauterivian, opening the door to the recognition of Valanginian permafrost in other polar and high-altitude paleolatitudes.

Major perturbations of the global carbon cycle during the Cretaceous have previously been attributed to increased levels of atmospheric CO$_2$ related to intensified global tectonic activity and/or widespread volcanic activity[70]. However, the effect of large igneous provinces (LIPs) seems to have a different correlation with the global variation of the palaeo-atmospheric pCO$_2$[71] and sea-surface temperature (SST) (°C)[72]. Moreover, the relative role of these controls has been

re-evaluated and an alternate trigger for increased fertilization of the oceans should have existed[73].

We hypothesize that global permafrost thaw during the Cretaceous released significant volumes of greenhouse gases to the atmosphere as well as dissolved organic carbon (DOC) and other nutrients into watersheds, and marine waters. Thaw of organic-rich permafrost increases carbon release and may affect aquatic systems through carbon and nutrient additions[5]. The contribution of permafrost thaw to the Cretaceous global C balance, including during oceanic anoxic events (OAE) will have to be determined in future research dealing with ocean–continental cryosphere coupling associated with events of cryosphere degradation in the aftermaths of supergreenhouse cold snaps.

Thaw of permafrost in plateau desert basins as that reported from the Cretaceous of China may constitute a neglected example of abrupt permafrost thaw[74], postdating a global cold snap in the Hauterivian (Fig. 7) with plateau permafrost in China coeval with marine cryospheric indicators in Svalbard[65], Alaska[67,75], and Australia[6]. Global thaw of permafrost after the cold snap likely released carbon as greenhouse gases (CO$_2$ and CH$_4$) due to microbial decomposition of carbon previously frozen in permafrost, an example of positive feedback and likely to amplify climate warming[76].

The development of plateau permafrost during the Hauterivian correlates with a global drop of SST (°C)[72]. Furthermore, its disappearance correlates with a global increase in atmospheric pCO$_2$ inferred from pedogenic carbonates[71], with a global shift towards heavier values of $\delta^{13}Cc$ (‰)[65], coinciding also with a rapid rise of global SST (°C)[72]. The global synchroneity (interval 132.5–128 Ma) of the positive carbon isotope event, the SST variation, and the rise of atmospheric pCO$_2$ postdating the terrestrial record of plateau permafrost collectively points to a strong coupling of the ocean–atmosphere system. It also suggests that the disappearance of cryospheric systems

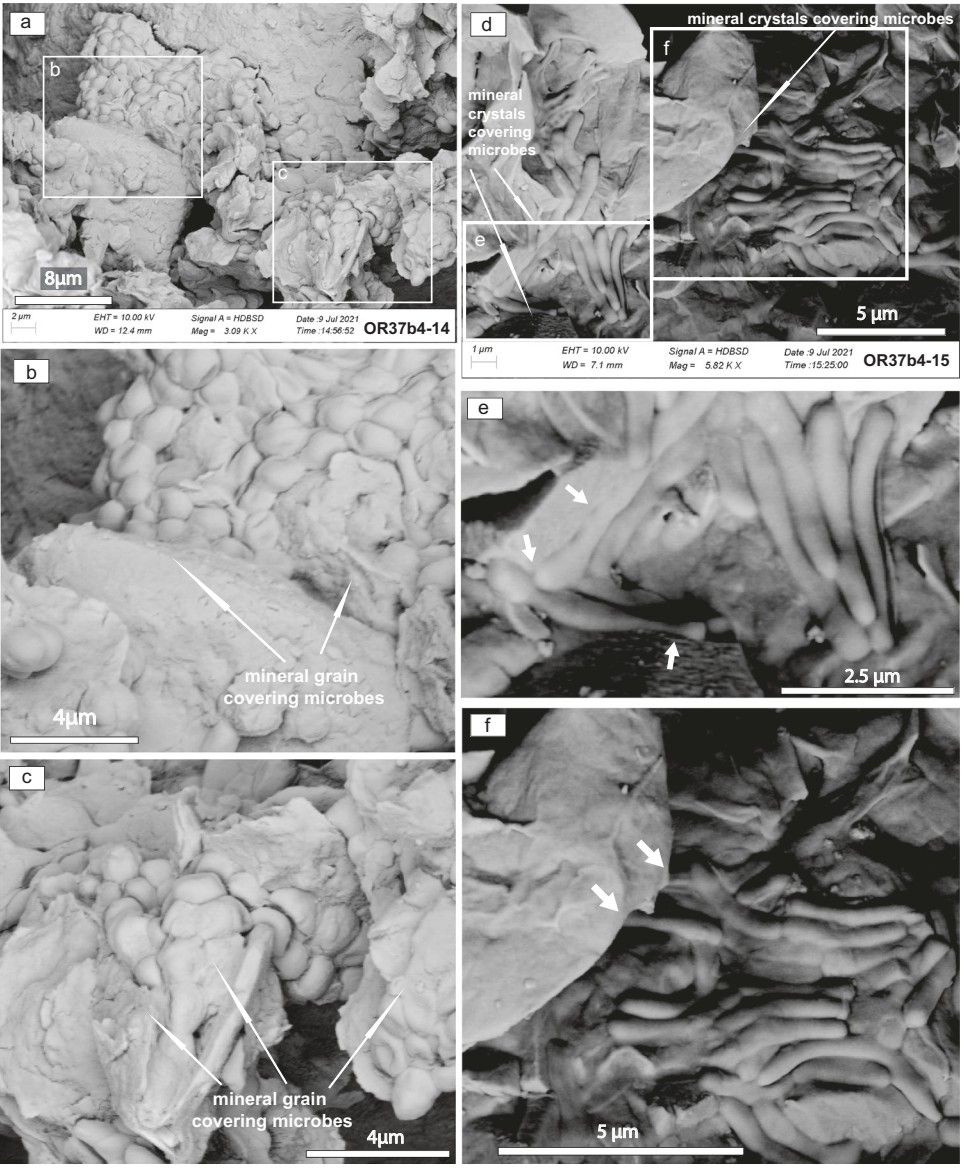

**Fig. 8 | Cretaceous permafrost microbiome. a** Sample OR37b4-14 shows mineralized fossil microorganisms gathered in a monolayer. The microbial cells are enveloped in sheaths and form a film, which contains numerous oval cells about 1.4 μm long and 1–1.2 μm wide. Rounded cells with diameters of about 1.4 μm and oval-flattened cells are also visible. The cells are attributed to eukaryotic microorganisms. See Supplementary Figs. 13, 14. **b** and **c** show mineralized microorganisms gathered in a monolayer. The microbial cells are enveloped in sheaths and form a biofilm, which contains numerous oval cells about 1.4 μm long and 1–1.2 μm wide. Rounded cells with a diameter of about 1.4 μm and oval-flattened cells are also visible. Probably these cells belong to eukaryotic microorganisms. Areas **b** and **c** demonstrate evidence of mineral grains covering microbial fossilized cells. **d** Sample OR37b4-15 shows the accumulation of rod-shaped bacterial forms 2.5–3.5 μm long and 0.4–0.5 μm wide. Rod-shaped cells are slightly curved with rounded ends and resemble Bacilli. See Supplementary Figs. 15, 16. **d** shows the accumulation of rod-shaped bacterial forms 2.5–3.5 μm long and 0.4–0.5 μm wide. Rod-shaped cells are slightly curved with rounded ends and resemble Bacilli. **e** and **f** mineral grains and cements cover fossil bacteria (white arrows).

associated with a global cold event may have affected the total exchangeable carbon reservoir.

Estimations made from mid-Pliocene (3.3–3.0 Ma) lacustrine systems from Tibet highlight that ca. 60% of alpine permafrost is vulnerable to thawing compared to ca. 20% of circumarctic permafrost[77]. These authors estimated that ca. 25% of permafrost carbon and the permafrost–climate feedback could arise from alpine areas[77].

At present, permafrost stores ~1600 BT of carbon[78] over 20% of Earth's terrestrial surface[62]. This store is nearly twice as large as the carbon stored today in the atmosphere[78]. We conclude that thaw of permafrost[76] after the Hautiverian cold snap promoted global warming of Cretaceous climates and fertilized the oceans. Antarctic evidence for glaciation of the South Pole during the Late Cretaceous[18] raises the

likelihood that permafrost occurred at that time in Antarctica. Further investigation on recent permafrost analogues and of Cretaceous oceanic deposits may shed light on the exact contribution of permafrost thaw[79] to the radiative forcing of global Cretaceous events.

## Methods

### Fieldwork and sample collection

During our field investigations from October 8th to 15th, 2020, we conducted sedimentological analysis of aeolian sandstones in the Luohe Fm at the Yucha Grand Canyon, Ganquan County, Central Ordos Basin. We focused on the sandstone wedge structures within the aeolian sandstones of a well-exposed outcrop, OR15 (Supplementary Fig. 1c and d). The Yucha Grand Canyon is located on the north side of

the Second Level Road of Ganquan County to Zhidan County, 40 km west of Ganquan County, and 72 km east of Zhidan County (Supplementary Fig. 1c and d). The entrance to the visitor center is in Zhangjiagou Village. From the visitor center, drive 3.7 km north to Yayaodi Village, and walk 366 m west to OR15 (Supplementary Fig. 1c and d).

As the outcrop OR15 with the wedges in the aeolian sandstone represents a steep cliff, we used a drone to investigate and photograph the upper portion of the outcrop. The drone model is the Mavic Air 2 (SZ DJI Technology Co., Ltd., Shenzhen, China).

For microscopy, samples from the wedges were collected with sterile tools and placed into sterile tightly sealed bags. In the laboratory, samples were ground up and prepared for microscopy inside a sterile box.

### Natural gamma ray (GR) logging data
The intensity of GR levels in the rock relates to its content of uranium (U), thorium (Th), and potassium (K), reflecting the amount of clay and organic matter[80]. Potassium is concentrated in common minerals such as clays, feldspar, mica, and chloride salts. U and Th are commonly found in minerals such as clays, heavy minerals, feldspars, and phosphate, whereas U is usually concentrated in organic matter[80]. The distribution of these radioactive elements, determined from GR data with the advantage of continuous and high resolution, has been used as a palaeoclimatic proxy for many cyclostratigraphic studies[81,82]. GR logs of Well Wuqi were selected for cyclostratigraphic analysis of the early Cretaceous in Ordos Basin (Supplementary Data 1 and 2).

### Time series analysis and modelling
Time series analysis of the GR data was performed with Acycle 2.0 software[83]. The long-term linear trend in GR data was removed by the detrending function. The $2\pi$ multitaper (MTM) method of spectral analysis with a red-noise null model was performed to detect significant frequency peaks. Sliding-window analysis with the evolutionary fast Fourier transform (FFT) method was used to examine changes in dominant frequency patterns. The Gaussian bandpass filter was used to filter 405-kyr signals. In addition, the correlation coefficient (COCO) analysis was applied to evaluate the optimal sedimentation rate for the studied sequence[83]. The La2004 orbital solution has been identified as a precise astronomical target for the period 134–129.4 Ma[84].

### Scanning electron microscope (SEM) analysis
The SEM instrument model was SIGMA300 (Carl Zeiss AG, Germany) and the whole experiment was carried out at the Institute of Multipurpose Utilization of Mineral Resources, Chinese Academy of Geological Sciences, Chengdu, China. A total of 9 samples for SEM analysis was collected. The sandstone wedge structures cropping out in the field were gently tapped into small blocks to obtain the natural section and then processed to a size of 1.5 cm*1.5 cm. The samples were fixed to the sample stage (1.3 cm*1.3 cm) by means of conductive adhesive and coated using the gold plating instrument with three sprays of 10 min each. Three samples were placed in the SEM sample compartment at a time. Sterile gloves and masks were worn throughout the process and the samples were cleaned repeatedly with compressed air to ensure they were not contaminated. The SEM was operated at 10 kV, HDBSD mode. The sample number was specified on the operating screen and the microbial phenomena were photographed at different magnifications after the optimum photographic definition was obtained by adjusting the axis distance, contrast, brightness, and focus. The energy dispersive X-ray spectroscopy energy spectrum was scanned as a point and surface scan, with the Au elements, removed and normalized to give the results. Photomicrographs were taken before and after the energy spectrum scan to observe any changes in microbial morphology and make sure the microbes observed were fossils and that no morphological changes occurred after the electron beam was fired.

## Data availability
The Gamma Ray log data of Wells Wuqi and Lingtai of the Cretaceous Luohe Fm. used in this study are available in the University of Sussex data Repository under accession code https://doi.org/10.25377/sussex.21610635.

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

## Acknowledgements
We thank Yuxiang Shi and Qiushuang Fan for their assistance in the field and helpful discussion. This work was jointly funded by the National Natural Science Foundation of China (Nos. 41872099, 42230310, 41888101, 91855213, 41602127) to Ch.W., Wq.T., C.M. This work is also funded by the *"Convocatoria de Ayudas para la recualificación del sistema universitario Español 2021–2023, Financiado por la Unión Europea-Next Generation EU", Vicerrectorado de Investigación, Universidad del País Vasco UPV/EHU* to J.P.R.L. This work is a contribution to the Research Group of the Basque Government IT-1602-22 (*Grupo Consolidado del Gobierno Vasco* IT-1602-22). This work is partially supported by the US Department of Energy, Office of Science, Office of Biological and Environmental Research, Genomic Science Programme under award number DE-SC0020369 to T.A.V. We are grateful to the PETROCHINA CHANGQING OILFIELD COMPANY for granting permission to use the subsurface data of the wells. We thank IPCC for granting permission for the use of Fig. 2.1 from ref. 44 (Hock et al., 2019). We thank Dr. Bernd Etzelmüller for the discussion of high-altitude permafrost, and Dr. Chris Burn for the discussion of permafrost patterned ground persisting beneath shallow lakes.

## Author contributions
J.P.R.L. and Ch.W. developed the scientific original idea. J.P.R.L. prepared the narrative of the main manuscript, sedimentological and architectural analyses, discussion, and figures, the Cretaceous palaeo-climates and global implications, the Supplementary Material (sedimentology description and discussion, permafrost wedge analysis description and discussions), prepared fossil figures and discussion, the West Himalayan analogue description and discussion, and its figures. Ch.W. gathered the fieldwork photographs, carried out the sampling, and the SEM analysis, contributed to the main manuscript, magnetostratigraphic correlation, geological setting, global implications, discussion, figure preparations and preparation of the Supplementary material and Methods sections. T.A.V. carried out the microbiological analysis of fossil bacteria and has contributed to the main manuscript, Supplementary material and figures. J.M. provided the discussion and analysis of recent analogue analysis and contributed to the main manuscript and Supplementary materials. Wq.T. carried out the astrochronological analysis of wells, developed the "Methods" section, and contributed to the Supplementary materials, as well as to the main manuscript and figures. C.M. carried out the comparative analysis of astronomical signals and palaeo-climatic events, Supplementary materials, and contributed to the main manuscript and figures.

## Competing interests
The authors declare no competing interests.
