## [Peer Review File · Nature Communications]

Reviewers' Comments:

Reviewer #1:

Remarks to the Author:

Review of Permafrost in the Cretaceous supergreenhouse (NCOMMS-22-29294-T) submitted to Nature Communications by Rodríguez-López et al.

This is a provocative, interesting, and well-written article that puts forth the concept of the existence of extensive permafrost during the (Early) Cretaceous supergreenhouse, and its thaw as an agent of carbon release that led to climate warming.

I have some suggestions for improvement that I outline below.

The authors seem to stay away from mentioning ocean anoxic events – why? Release of sufficient C (certainly from LIPs) are discussed as the causal agent of OAEs in existing literature. The authors suggest that C release from permafrost thaw is also an important agent of C release sufficient to cause climate warming, but do not get into discussing any OAEs or C isotope excursions except for mention of a lack of excellent time correlation with the Weissert event (in the Valanginian). I think that it should be acknowledged then, that the early Hauterivian C release proposed from permafrost thaw was apparently (?) insufficient to induce an OAE and C isotope excursion.

LIPs – can the authors name the LIPs on their Figure 4. I make a specific comment below about the HALIP. The description about mechanisms of how LIPs affect paleoclimate change is presented overly simplistically in the manuscript, but I understand there are word limits and abundant literature on how the construction and deconstruction of one or more LIPs can affect global climate and the C cycle. Still, I think adding a sentence or two would be helpful, given that the premise of the article is that permafrost thaw and associated C release was at least as large (or larger) an agent of climate change as LIPs were.

Valanginian glendonites – this is ignored in the manuscript. The widespread occurrence of glendonites in Valanginian-aged strata in the Canadian Arctic (Deer Bay Fm) should be acknowledged in the manuscript and Fig. 4.

Para 35 – the authors do not mention glendonites as indicators of polar ice. See Grasby et al. 2017 and references therein. McArthur et al. 2007 use isotopes to reconstruct polar ice volume and paleotemperatures. Price (1999) also suggests polar ice.

Grasby, S.E., McCune, G.E., Beauchamp, B., Galloway, J.M., 2017. Lower Cretaceous cold snaps led to widespread glendonite occurrences in the Sverdrup Basin, Canadian High Arctic. GSA Bulletin 129, 771-787.

McArthur, J.M., Janssen, N.M.M., Reboulet, S., Leng, M.J., Thirlwall, M.V., van de Schootbrugge, B., 2007. Palaeotemperatures, polar ice-volume, and isotope stratigraphy (Mg/Ca, d18O, d13C, 87Sr/86Sr): The Early Cretaceous (Berriasian, Valanginian, Hauterivian). Palaeogeography, Palaeoclimatology, Palaeoecology 248, 391e430.

Price, G.D., 1999, The evidence and implications of polar ice during the Mesozoic: Earth-Science Reviews, v. 48, p. 183–210, doi: 10.1016/S0012-8252(99)00048-3.

Para 40 – remove of in last line (understanding climate dynamics)

Para 75 – lower case l for late Pleistocene

Para 75 – NWT is NT

Para 95 – the authors claim that they present the “first direct evidence for Mesozoic ice on Earth” but I consider geochemical evidence and ice rafted debris to be direct evidence. I suggest rewording to the first direct sedimentological evidence of Mesozoic ice on Earth, or changing it to

be ground ice, as the previous studies using isotopes and IRD were reconstructing polar ice in the Arctic Ocean.

Para 125 – the authors compare the astrochronologically calibrated age of the sandstone wedges from Aeolian desert systems in China to the occurrence of Hauterivian glendonites in Svalbard – but not the Valanginian-aged glendonites in the Deer Bay Formation of Arctic Canada (Grasby et al., 2017; Galloway et al., 2020). From my reading of the manuscript, the age of the aeolian wedges seem to accommodate a Valanginian age?

Galloway, J.M., Vickers, M., Price, G.D., Poulton, T.P., Grasby, S.E., Hadlari, T., Beauchamp, B., Sulphur, K. 2020. Finding the VOICE: Organic carbon isotope chemostratigraphy of the Late Jurassic-Early Cretaceous of Arctic Canada. *Geological Magazine* 157: 1643-1657.

Para 140 – High Arctic – capital H. Where in the High Arctic? Canadian High Arctic? Elsewhere in the sentence the localities are listed (Alaska, Canada, Svalbard).

Para 450 – Figure 4. Can the authors name the LIPs (in the figure caption?) The HALIP extends from 122 +/- 22 (Dockman et al., 2016) and is an episodic and protracted LIP that extends into the Late Cretaceous (to ~90 to 85 Ma) so depicting a LIP (if this is HALIP) as terminating at 116 Ma provides an incomplete depiction of that LIP.

Fig. 2 – there are glendonites in Valanginian strata of the Deer Bay Fm in the Canadian Arctic (Sverdrup Basin). This cannot be ignored. See Grasby et al. 2017 and Galloway et al. 2020.

Overall, I recommend publishing of this submission after revisions. It was a very interesting read. I am not an expert of permafrost structures and the expertise of someone in that area should be sought.

Additional editorial comments are in the annotated word document attached to my review.

My identity may be revealed – Dr. Jennifer Galloway, Geological Survey of Canada, Calgary, Alberta.

Reviewer #2:

Remarks to the Author:

Review

This paper reconstructs permafrost in the Himalaya in deep time making finally some inferences that the permafrost thaw might have an impact on CO₂ and temperature, drawing on something potentially similar in the Pliocene.

I am a carbon cycle modeller and I cannot comment on the data itself. However, what I can say is that the impact of permafrost thaw on atmospheric CO₂ in terms of CO₂ release is an experiment very similar to modern fossil fuel emissions and for that a multitude of simulation studies exists. The key messages of all studies is, that the amount of CO₂, that stays in the atmosphere (the so-called airborne fraction) and might therefore change CO₂ partial pressure and global mean surface temperature is decaying pretty fast with time. First, the ocean absorbs CO₂ more than halving the atmospheric CO₂ anomaly, then resulting changes in deep ocean carbonate chemistry lead to a carbonate compensation feedback (dissolution of sediments) before finally continental weathering brings CO₂ back to initial values. Typical numbers of this airborne fraction here are 0.4 on millennial time scales (ocean), going down to 0.1 on 10-100kyr (sediments), and back to initial values after latest 1 Myr. This would imply that the 1600 PgC in present day permafrost if completely emitted to the atmosphere would lead to an anomaly in atmospheric carbon of +640 PgC (~ +300 ppm) after some thousand years, and to only +160 PgC (~ +75 ppm) after 10-100ky, and zero anomaly after latest 1 Ma. In other words, such a signal would not show up at all in the CO₂ record shown in Fig 4 which seemed to have a resolution of ~1 Ma. These numbers are on the order of 2x CO₂ (or below), implying that they might be responsible for a short-term (kyr)

global mean surface warming of 2.6-3.9 K using most recent estimates on climate sensitivity (Sherwood et al., 2020). These are maximum estimates to give an idea how big an impact might be.

Therefore, what is inferred from this study seems to be rather irrelevant for CO₂ on the available resolution of the data. Permafrost thaw can only be responsible for rather short-term (and relatively small) peaks in CO₂. Thus, the invoked effect that permafrost, instead of volcanism and/or tectonic might be responsible for major carbon cycle change in the Cretaceous is in my view not supported.

CH₄ emitted from thawing permafrost typically makes up only some percentages of the emitted gases. Although the radiative forcing potential of CH₄ is about 20-100 times higher than that of CO₂, it is even more a short term effect and cannot compensate for a missing warming not contained in CO₂.

References:

Long-term effect of CO₂ emissions on C cycle:

Joos, F., Roth, R., Fuglestedt, J. S., Peters, G. P., Enting, I. G., von Bloh, W., Brovkin, V., Burke, E. J., Eby, M., Edwards, N. R., Friedrich, T., Frölicher, T. L., Halloran, P. R., Holden, P. B., Jones, C., Kleinen, T., Mackenzie, F. T., Matsumoto, K., Meinshausen, M., Plattner, G.-K., Reisinger, A., Segschneider, J., Shaffer, G., Steinacher, M., Strassmann, K., Tanaka, K., Timmermann, A., and Weaver, A. J.: Carbon dioxide and climate impulse response functions for the computation of greenhouse gas metrics: a multi-model analysis, *Atmos. Chem. Phys.*, 13, 2793–2825, <https://doi.org/10.5194/acp-13-2793-2013>, 2013

Colbourn, G., Ridgwell, A., and Lenton, T. M. (2015). The time scale of the silicate weathering negative feedback on atmospheric CO₂. *Glob. Biogeochem. Cycles* 29, 583–596. doi: 10.1002/2014GB005054

Köhler P (2020) Anthropogenic CO₂ of High Emission Scenario Compensated After 3500 Years of Ocean Alkalinization With an Annually Constant Dissolution of 5 Pg of Olivine. *Front. Clim.* 2:575744. doi: 10.3389/fclim.2020.575744

CH₄ in permafrost thaw:

Schuur, E. et al. Expert assessment of vulnerability of permafrost carbon to climate change. *Clim. Change* 119, 359–374 (2013).

Climate sensitivity

Sherwood, S. C., Webb, M. J., Annan, J. D., Armour, K. C., Forster, P. M., Hargreaves, J. C., et al. (2020). An assessment of Earth's climate sensitivity using multiple lines of evidence. *Reviews of Geophysics*, 58, e2019RG000678. <https://doi.org/10.1029/2019RG000678>

Reviewer #3:

Remarks to the Author:

The paper "Permafrost in the Cretaceous supergreenhouse" by Rodríguez-López et al. provides sedimentological evidence of ground ice in large dune systems on plateau deserts of China during the early Cretaceous some 132 million years ago. Permafrost developed in a cold spell during a time of overall Cretaceous greenhouse warmth. The authors also found traces of microbial activity, typically encountered in environments of thawing frozen ground. They postulate that cryogenic

processes released biogenic trace gases that might have contributed to the carbon cycle and the support of strong greenhouse conditions.

The study is based on a sophisticated methodological approach. The most important step was the sedimentological recognition and perfect interpretation of cryogenic wedge structures in the field. Preserved and lithified signs of microbial life were checked by SEM analyses. The inference of paleoclimatic parameters is convincing.

The text is well organized, clearly written and illustrated. Adequate references support the discussion.

The identification of times with ice presence during the Cretaceous is not a novel message, but usually was only assigned to glacial ice on land and sea. This article provides strong clues for former permafrost conditions in highlands of the low latitudes, which under young ice-house conditions show analogues in Alaska and the Himalayas. Comparison with climate dynamics in the Pliocene also underline the significance of the interpretations. The finding of Cretaceous permafrost is unique and worth publishing at high scientific level. The implication of feedback loops in carbon cycling stimulates a fundamental hypothesis for further studies.

Comments and suggested improvements

- The title is very general and suggests both super greenhouse and permafrost conditions through the Cretaceous, which is known as a long-lasting greenhouse period. The mid-Cretaceous apparently was hottest and by some authors is referred to as a supergreenhouse stage. However, this work relates to the lower Cretaceous. Permafrost is only postulated for a "cold snap" of a few hundred kiloyears, which is consistent with reported cooling and glacial features worldwide. Terminology and timing should be dealt with attention.
- Line 95: "We present the first direct evidence of Mesozoic ice on Earth." What do you mean with direct evidence? Other evidence on glacial ice is known from the literature. Your evidence is also indirect. Reference 1 also claims cold temperatures in your study area. Please clarify the findings of this paper in relation to your article.
- Astrochronology has been applied to aeolian sequences of the Luohe Formation. Please indicate the quality of the depositional units in relation to unconformities and the degree of temporal preservation of the sediment record.
- The idea of microbial carbon release from permafrost as a driver of Cretaceous variations in atmospheric carbon is reasonable, but should be substantiated by further arguments. Today, there is a big pool of soil carbon in permafrost regions, as correctly cited. How was the situation in the Cretaceous? The studied section in China demonstrates a desert environment, affected by permafrost in sandy substrates with low organic matter. What about the degradation of fossil lacustrine deposits and palaeosols. Further studies have to check the soil inventories in the early Cretaceous greenhouse and to calculate the source-sink relationships in the carbon cycle at that time.

Response to reviewers:

REVIEWER COMMENTS

Reviewer #1 (Remarks to the Author):

Review of Permafrost in the Cretaceous supergreenhouse (NCOMMS-22-29294-T) submitted to Nature Communications by Rodríguez-López et al.

This is a provocative, interesting, and well-written article that puts forth the concept of the existence of extensive permafrost during the (Early) Cretaceous supergreenhouse, and its thaw as an agent of carbon release that led to climate warming.

I have some suggestions for improvement that I outline below.

The authors seem to stay away from mentioning ocean anoxic events – why? Release of sufficient C (certainly from LIPs) are discussed as the causal agent of OAEs in existing literature. The authors suggest that C release from permafrost thaw is also an important agent of C release sufficient to cause climate warming, but do not get into discussing any OAEs or C isotope excursions except for mention of a lack of excellent time correlation with the Weissert event (in the Valanginian). I think that it should be acknowledged then, that the early Hauterivian C release proposed from permafrost thaw was apparently (?) insufficient to induce an OAE and C isotope excursion.

Response:

Thank you for this interesting comment. We have not included information regarding Cretaceous Oceanic Anoxic Events (OAE) indeed, because this is a further step on our research group: we are now analysing the potential contribution of permafrost thaw-derived C, as well as to quantify the release of CH₄ and CO₂ from the recent analogue in Western Himalayas.

This paper constitutes the foundations for further research papers in progress.

Yes, we agree. We mentioned the Weisser event, because this is the one closest, in time, with respect to the Hauterivian permafrost identified in China.

LIPs – can the authors name the LIPs on their Figure 4. I make a specific comment below about the HALIP. The description about mechanisms of how LIPs affect paleoclimate change is presented overly simplistically in the manuscript, but I understand there are word limits and abundant literature on how the construction and deconstruction of one or more LIPs can affect global climate and the C cycle. Still, I think adding a sentence or two would be helpful, given that the premise of the article is that permafrost thaw and associated C release was at least as large (or larger) an agent of climate change as LIPs were.

Response:

The information provided about LIPs in Fig. 4 is from the reference Li, X. et al., (2014):

(Their Fig. 8: LIP 1: 134–129 Ma, Paraná and Etendeka traps (Janasi, Freitas & Heaman, 2011) and Comei-Bunbury LIP (Zhu et al. 2009); LIP 2: 126–122 Ma, Ontong

Java Plateau, Manihiki Plateau and Hikurangi Plateau (summary by Kuroda et al. 2011); LIP 3: 120–116 Ma, Kerguelen Plateau–Rajmahal traps (Wignall, 2001; Courtillot & Renne, 2003).

Yes, we have added the names of the LIPs in the Figure caption following the recommendation of Reviewer #1.

Following the recommendation of the Editor we modified the sentences dealing with the comparison of the C contribution from permafrost thaw with the once from LIPs.

This quantification will have to be carried out in further publications, once we have recognized the existence of Cretaceous permafrost.

Following the indications of Reviewer #1 we have included the following sentence dealing with the OAE in the discussion section:

The contribution of permafrost thaw to the Cretaceous global C balance, including during Oceanic Anoxic Events (OAE) will have to be determined in future research dealing with the ocean-continental cryosphere coupling associated with events of cryosphere degradation in the aftermaths of supergreenhouse cold snaps.

We previously published the possible role of HALIP on volcanic-ice interactions leading to extreme icebergs reaching tropical paleolatitudes (Rodríguez-López et al., 2016, *Palaeogeography, Paleoclimate, Palaeoecology*).

Valanginian glendonites – this is ignored in the manuscript. The widespread occurrence of glendonites in Valanginian-aged strata in the Canadian Arctic (Deer Bay Fm) should be acknowledged in the manuscript and Fig. 4.

Response:

Thanks for pointing this out. We consider it important to add the information about the Valanginian glendonites from Canada that will be a starting point to recognize other possible early Cretaceous permafrost system during the Valanginian.

We have also added the following paragraph in the second paragraph of the Discussion section:

“The occurrence of well-known Valanginian-age ice-rafted debris (Galloway et al., 2020) opens the door to the recognition of Valanginian permafrost in other polar and high-altitude paleolatitudes”.

Para 35 – the authors do not mention glendonites as indicators of polar ice. See Grasby et al. 2017 and references therein. McArthur et al. 2007 use isotopes to reconstruct polar ice volume and paleotemperatures. Price (1999) also suggests polar ice.

Grasby, S.E., McCune, G.E., Beauchamp, B., Galloway, J.M., 2017. Lower Cretaceous cold snaps led to widespread glendonite occurrences in the Sverdrup Basin, Canadian High Arctic. *GSA Bulletin* 129, 771-787.

McArthur, J.M., Janssen, N.M.M., Reboulet, S., Leng, M.J., Thirlwall, M.V., van de Schootbrugge, B., 2007. Palaeotemperatures, polar ice-volume, and isotope stratigraphy (Mg/Ca, d18O, d13C, 87Sr/86Sr): The Early Cretaceous (Berriasian, Valanginian, Hauterivian). *Palaeogeography, Palaeoclimatology, Palaeoecology* 248, 391e430.

Price, G.D., 1999, The evidence and implications of polar ice during the Mesozoic: *Earth-Science Reviews*, v. 48, p. 183–210, doi: 10.1016/S0012-8252(99)00048-3.

Response:

Done.

We have added the suggested references by Reviewer #1:

Grasby, S.E., McCune, G.E., Beauchamp, B., Galloway, J.M., 2017. Lower Cretaceous cold snaps led to widespread glendonite occurrences in the Sverdrup Basin, Canadian High Arctic. *GSA Bulletin* 129, 771-787.

McArthur, J.M., Janssen, N.M.M., Reboulet, S., Leng, M.J., Thirlwall, M.V., van de Schootbrugge, B., 2007. Palaeotemperatures, polar ice-volume, and isotope stratigraphy (Mg/Ca, d18O, d13C, 87Sr/86Sr): The Early Cretaceous (Berriasian, Valanginian, Hauterivian). *Palaeogeography, Palaeoclimatology, Palaeoecology* 248, 391e430.

Price, G.D., 1999, The evidence and implications of polar ice during the Mesozoic: *Earth-Science Reviews*, v. 48, p. 183–210, doi: 10.1016/S0012-8252(99)00048-3.

Para 40 – remove of in last line (understanding climate dynamics)

Response:

Done.

Para 75 – lower case l for late Pleistocene

Response:

Done.

Para 75 – NWT is NT

Response:

Done.

Para 95 – the authors claim that they present the “first direct evidence for Mesozoic ice on Earth” but I consider geochemical evidence and ice rafted debris to be direct evidence. I suggest rewording to the first direct sedimentological evidence of Mesozoic ice on Earth, or changing it to be ground ice, as the previous studies using isotopes and IRD were reconstructing polar ice in the Arctic Ocean.

Response:

Done.

We agree with both Reviewers #1 and #3 and we deleted the word “direct”.

Para 125 – the authors compare the astrochronologically calibrated age of the sandstone wedges from Aeolian desert systems in China to the occurrence of Hauterivian glendonites in Svalbard – but not the Valanginian-aged glendonites in the Deer Bay Formation of Arctic Canada (Grasby et al., 2017; Galloway et al., 2020). From my reading of the manuscript, the age of the aeolian wedges seem to accommodate a Valanginian age?

Galloway, J.M., Vickers, M., Price, G.D., Poulton, T.P., Grasby, S.E., Hadlari, T., Beauchamp, B., Sulphur, K. 2020. Finding the VOICE: Organic carbon isotope chemostratigraphy of the Late Jurassic-Early Cretaceous of Arctic Canada. *Geological Magazine* 157: 1643-1657.

Response:

Done.

The obtained astrochronological age obtained for the interval bearing the permafrost wedges is ca. 132.49–132.17 Ma (Hauterivian 132.6 – 129.4 Ma), and we consider very important to add the reference to Valanginian-aged glendonites proposed by Reviewer #1. This will help also to search for other possible Valanginian permafrost systems in the rock record worldwide.

We have added the reference proposed by Reviewer #1:

Galloway, J.M., Vickers, M., Price, G.D., Poulton, T.P., Grasby, S.E., Hadlari, T., Beauchamp, B., Sulphur, K. 2020. Finding the VOICE: Organic carbon isotope chemostratigraphy of the Late Jurassic-Early Cretaceous of Arctic Canada. *Geological Magazine* 157: 1643-1657.

We have also added the following paragraph in the second paragraph of the Discussion section:

“The occurrence of well-known Valanginian-age ice-rafted debris (Galloway et al., 2020) opens the door to the recognition of Valanginian permafrost in other polar and high-altitude paleolatitudes”.

Para 140 – High Arctic – capital H. Where in the High Arctic? Canadian High Arctic? Elsewhere in the sentence the localities are listed (Alaska, Canada, Svalbard).

Response:

Done.

We have added “Svalbard” as suggested by Reviewer #1.

Para 450 – Figure 4. Can the authors name the LIPs (in the figure caption?) The HALIP extends from 122 +/- 22 (Dockman et al., 2016) and is an episodic and protracted LIP that extends into

the Late Cretaceous (to ~90 to 85 Ma) so depicting a LIP (if this is HALIP) as terminating at 116 Ma provides an incomplete depiction of that LIP.

Fig. 2 – there are glendonites in Valanginian strata of the Deer Bay Fm in the Canadian Arctic (Sverdrup Basin). This cannot be ignored. See Grasby et al. 2017 and Galloway et al. 2020.

Response:

Done.

We have modified Fig. 2 following the indications of Reviewer #1: We have included in Fig. 2 the data regarding Valanginian glendonites and dropstones in the Canadian Arctic, adding the following references suggested by Reviewer #1:

Galloway, J.M., Vickers, M., Price, G.D., Poulton, T.P., Grasby, S.E., Hadlari, T., Beauchamp, B., Sulphur, K. 2020. Finding the VOICE: Organic carbon isotope chemostratigraphy of the Late Jurassic-Early Cretaceous of Arctic Canada. *Geological Magazine* 157: 1643-1657.

Grasby, S.E., McCune, G.E., Beauchamp, B., Galloway, J.M., 2017. Lower Cretaceous cold snaps led to widespread glendonite occurrences in the Sverdrup Basin, Canadian High Arctic. *GSA Bulletin* 129, 771-787.

Additionally, we have added the occurrence of dropstones in the Deer Bay Fm. reported in the following recent paper:

Schneider, S. et al., 2020, Macrofauna and biostratigraphy of the Rollrock Section, northern Ellesmere Island, Canadian Arctic Islands e a comprehensive high latitude archive of the Jurassic-Cretaceous transition. *Cretaceous Research*, 114, 104508.

We have also included the data on these Valanginian dropstones and glendonites in the Introduction, and the Discussion as well as the figure caption of Fig. 2.

Overall, I recommend publishing of this submission after revisions. It was a very interesting read. I am not an expert of permafrost structures and the expertise of someone in that area should be sought.

Additional editorial comments are in the annotated word document attached to my review.

My identity may be revealed – Dr. Jennifer Galloway, Geological Survey of Canada, Calgary, Alberta.

We are grateful to Dr. Jennifer Galloway for her comments and suggestions.

Reviewer #2 (Remarks to the Author):

Review

This paper reconstructs permafrost in the Himalaya in deep time making finally some inferences that the permafrost thaw might have an impact on CO2 and temperature, drawing on something

potentially similar in the Pliocene.

I am a carbon cycle modeller and I cannot comment on the data itself. However, what I can say is that the impact of permafrost thaw on atmospheric CO₂ in terms of CO₂ release is an experiment very similar to modern fossil fuel emissions and for that a multitude of simulation studies exists. The key messages of all studies is, that the amount of CO₂, that stays in the atmosphere (the so-called airborne fraction) and might therefore change CO₂ partial pressure and global mean surface temperature is decaying pretty fast with time. First, the ocean absorbs CO₂ more than halving the atmospheric CO₂ anomaly, then resulting changes in deep ocean carbonate chemistry lead to a carbonate compensation feedback (dissolution of sediments) before finally continental weathering brings CO₂ back to initial values. Typical numbers of this airborne fraction here are 0.4 on millennial time scales (ocean), going down to 0.1 on 10-100kyr (sediments), and back to initial values after latest 1 Myr. This would imply that the 1600 PgC in present day permafrost if completely emitted to the atmosphere would lead to an anomaly in atmospheric carbon of +640 PgC (~ +300 ppm) after some thousand years, and to only +160 PgC (~ +75 ppm) after 10-100ky, and zero anomaly after latest 1 Ma. In other words, such a signal would not show up at all in the CO₂ record shown in Fig 4 which seemed to have a resolution of ~1 Ma. These numbers are on the order of 2x CO₂ (or below), implying that they might be responsible for a short-term (kyr) global mean surface warming of 2.6-3.9 K using most recent estimates on climate sensitivity (Sherwood et al., 2020). These are maximum estimates to give an idea how big an impact might be.

Therefore, what is inferred from this study seems to be rather irrelevant for CO₂ on the available resolution of the data. Permafrost thaw can only be responsible for rather short-term (and relatively small) peaks in CO₂. Thus, the invoked effect that permafrost, instead of volcanism and/or tectonic might be responsible for major carbon cycle change in the Cretaceous is in my view not supported.

CH₄ emitted from thawing permafrost typically makes up only some percentages of the emitted gases. Although the radiative forcing potential of CH₄ is about 20-100 times higher than that of CO₂, it is even more a short term effect and cannot compensate for a missing warming not contained in CO₂.

Response to Reviewer:

We thank the reviewer for their valuable comments. Accordingly, and as mentioned above, we have toned down the comparison of permafrost thaw with volcanism as a major geological source of carbon.

We note that the understanding of supergreenhouse cryosphere in deep time is in its infancy, in particular the Cretaceous and Eocene supergreenhouses. This manuscript constitutes the foundations for further investigations and analysis.

This forcing mechanism of cryosphere degradation must be investigated further, and we think that the detail analysis of oceanic series coeval with events of degradation of continental permafrost will help us to understand better the real impact of permafrost thaw in deep time palaeoclimates.

References:

Long-term effect of CO₂ emissions on C cycle:

Joos, F., Roth, R., Fuglestedt, J. S., Peters, G. P., Enting, I. G., von Bloh, W., Brovkin, V., Burke, E. J., Eby, M., Edwards, N. R., Friedrich, T., Frölicher, T. L., Halloran, P. R., Holden, P. B., Jones, C., Kleinen, T., Mackenzie, F. T., Matsumoto, K., Meinshausen, M., Plattner, G.-K., Reisinger, A., Segschneider, J., Shaffer, G., Steinacher, M., Strassmann, K., Tanaka, K., Timmermann, A., and Weaver, A. J.: Carbon dioxide and climate impulse response functions for

the computation of greenhouse gas metrics: a multi-model analysis, *Atmos. Chem. Phys.*, 13, 2793–2825, <https://doi.org/10.5194/acp-13-2793-2013>, 2013

Colbourn, G., Ridgwell, A., and Lenton, T. M. (2015). The time scale of the silicate weathering negative feedback on atmospheric CO₂. *Glob. Biogeochem. Cycles* 29, 583–596. doi: 10.1002/2014GB005054

Köhler P (2020) Anthropogenic CO₂ of High Emission Scenario Compensated After 3500 Years of Ocean Alkalinization With an Annually Constant Dissolution of 5 Pg of Olivine. *Front. Clim.* 2:575744. doi: 10.3389/fclim.2020.575744

CH₄ in permafrost thaw:

Schuur, E. et al. Expert assessment of vulnerability of permafrost carbon to climate change. *Clim. Change* 119, 359–374 (2013).

Climate sensitivity

Sherwood, S. C., Webb, M. J., Annan, J. D., Armour, K. C., Forster, P. M., Hargreaves, J. C., et al. (2020). An assessment of Earth's climate sensitivity using multiple lines of evidence. *Reviews of Geophysics*, 58, e2019RG000678. <https://doi.org/10.1029/2019RG000678>

We thank Reviewer #2 for the references; as also suggested by the Editor, we have toned it down deleting the direct comparison of permafrost thaw-derived C with those from volcanism and so we do not add more discussion on this topic in the manuscript.

We are grateful to Reviewer #2 for the comments and suggestions.

Reviewer #3 (Remarks to the Author):

The paper “Permafrost in the Cretaceous supergreenhouse” by Rodríguez-López et al. provides sedimentological evidence of ground ice in large dune systems on plateau deserts of China during the early Cretaceous some 132 million years ago. Permafrost developed in a cold spell during a time of overall Cretaceous greenhouse warmth. The authors also found traces of microbial activity, typically encountered in environments of thawing frozen ground. They postulate that cryogenic processes released biogenic trace gases that might have contributed to the carbon cycle and the support of strong greenhouse conditions.

The study is based on a sophisticated methodological approach. The most important step was the sedimentological recognition and perfect interpretation of cryogenic wedge structures in the field. Preserved and lithified signs of microbial life were checked by SEM analyses. The inference of paleoclimatic parameters is convincing.

The text is well organized, clearly written and illustrated. Adequate references support the discussion.

The identification of times with ice presence during the Cretaceous is not a novel message, but usually was only assigned to glacial ice on land and sea. This article provides strong clues for former permafrost conditions in highlands of the low latitudes, which under young ice-house conditions show analogues in Alaska and the Himalayas. Comparison with climate dynamics in

the Pliocene also underline the significance of the interpretations. The finding of Cretaceous permafrost is unique and worth publishing at high scientific level. The implication of feedback loops in carbon cycling stimulates a fundamental hypothesis for further studies.

Comments and suggested improvements

- The title is very general and suggests both super greenhouse and permafrost conditions through the Cretaceous, which is known as a long-lasting greenhouse period. The mid-Cretaceous apparently was hottest and by some authors is referred to as a supergreenhouse stage. However, this work relates to the lower Cretaceous. Permafrost is only postulated for a “cold snap” of a few hundred kiloyears, which is consistent with reported cooling and glacial features worldwide. Terminology and timing should be dealt with attention.

Response:

Thanks very much. The proposed title “Permafrost in the Cretaceous supergreenhouse” could be replaced by “Permafrost in the Cretaceous supergreenhouse and its Himalayan analogue”.

- Line 95: “We present the first direct evidence of Mesozoic ice on Earth.” What do you mean with direct evidence? Other evidence on glacial ice is known from the literature. Your evidence is also indirect. Reference 1 also claims cold temperatures in your study area. Please clarify the findings of this paper in relation to your article.

Response:

As this has been also indicated by Reviewer #1, we have deleted the word “direct” from the text, and regarding the other references, this is from another research group suggesting cold evidence in China but not presenting hard evidence neither analogues nor global implications. We have recently summarized the up-to-date data on the Cretaceous cryospheres in China in Wu, Rodríguez-López and Santosh (2022) *Geoscience Frontiers*.

Our results corroborate the Cretaceous models of paleotemperatures by the palaeotemperatures models for the Valanginian–Hauterivian stages (Lunt et al., 2016) that predicted an annual mean surface air temperature (at 1.5 m) for the Ordos Basin plateau desert basin $\leq 0^{\circ}\text{C}$ and temperatures -8°C for the bounding desert mountain ranges.

- Astrochronology has been applied to aeolian sequences of the Luohe Formation. Please indicate the quality of the depositional units in relation to unconformities and the degree of temporal preservation of the sediment record.

Response:

The sedimentary record of the Luohe Fm and other Cretaceous aeolian desert depositional systems is exceptional (cf., Wu, Rodríguez-López and Santosh, 2022, *Geoscience Frontiers*, 13, 101454.).

Both aeolian supersurfaces are main genetic limits indicating drastic changes in the depositional system (cf., Rodríguez-López et al., 2013, *Aeolian Research*). The genesis of these supersurfaces was similar to the recurrent processes of destruction due to rise of phreatic level recorded in the Western Himalayas analogue.

We have added the following paragraphs in the first Results section of the manuscript:

The preservation potential of the Cretaceous desert aeolian dune fields (ergs) is exceptional associated with the temporal and spatial evolution of subsidence rate associated with the Tethys and Pacific subduction plates in southeast Eurasia (Wu et al., 2022).

The temporal preservation of the sedimentary record of the permafrost wedges by comparison with recent and Quaternary analogues can represent

- The idea of microbial carbon release from permafrost as a driver of Cretaceous variations in atmospheric carbon is reasonable, but should be substantiated by further arguments. Today, there is a big pool of soil carbon in permafrost regions, as correctly cited. How was the situation in the Cretaceous? The studied section in China demonstrates a desert environment, affected by permafrost in sandy substrates with low organic matter. What about the degradation of fossil lacustrine deposits and palaeosols. Further studies have to check the soil inventories in the early Cretaceous greenhouse and to calculate the source-sink relationships in the carbon cycle at that time.

Response:

This is a very interesting question, thank you.

As mentioned before, this first paper presents the foundations for the analysis and recognition of continental cryosphere during the supergreenhouse Cretaceous.

As mentioned by Reviewer #3, the occurrence of fossil bacteria as occur in Recent thawing permafrost in the Arctic opens new questions and stimulates further studies and research; we do agree that a soil inventory and quantification of permafrost C pools is needed.

We are working on the quantification of C pools on the Recent analogue from Himalayas in our project as well as on the geochemical evidence from Cretaceous oceanic series.

We think that our manuscript will stimulate other colleagues to search for further evidence worldwide.

Permafrost features are very probably common in Lower Cretaceous successions in the Arctic and Antarctica and we encourage colleagues to look for possible neglected evidences of permafrost in the paleo-poles. This will help to characterize the Cretaceous cryosphere microbiome which, until this manuscript, was not even considered to exist.

We are grateful to Reviewer #3 for the comments and suggestions.

Reviewers' Comments:

Reviewer #1:

Remarks to the Author:

This is my second review of this manuscript. The authors sufficiently addressed the comments and concerns of the reviewers in their response letter. I will only make a few small comments on their rebuttal to my suggestions. First, I made an error - the reference I suggested they cite was Galloway et al., 2019 (not 2020). I recommend the authors add the HALIP to Figure 9 as well. And lastly, please do not cite Galloway et al. 2019 for ice rafted debris - its not a primary reference for this. Grasby et al. 2017 can also be cited for Valanginian glendonite occurrences in the Canadian Arctic. Other than these comments my original review comments remain: the article is well-written, provocative, and should be published in Nature Communications. Sincerely, Jennifer Galloway, Geological Survey of Canada.

Reviewer #2:

Remarks to the Author:

Review on Permafrost in the Cretaceous supergreenhouse

from

Rodríguez-López et al
submitted to Nature Communications
NCOMMS-22-29294A

I am reviewer #2 from the first round of reviews.

The authors followed my main concern of taken out permafrost thaw as an alternative to other long-term carbon cycle processes.

My initial suggestions in the last round of reviews was, that once this main claim is not supported anymore the paper is probably rejected, which is why I did not spend too much time on details.

However, other reviewers and the editorial board seemed to be more positive on the paper at large. Therefore, I try to fill in what I think is still incorrect on the details in the discussion around Figure 9.

- I suggest to replace numbers for addressing the subfigures in Fig 9 by letters, (Fig 9 1-12 -> Fig 9 a-l)

- CO2 plotted in Fig 9 is not based on the best sources in the literature:

- For the Cretaceous (Fig 9-2) a record was chosen that shows a rapid decline at 132Ma, while the review by Foster et al (2017) DOI: 10.1038/ncomms14845 shows a much more diverse picture. Based on Foster I would argue that you cannot claim any large changes in CO2 happening at that point in time.

- For the Pliocene (Fig 9-8) the authors plot CO2 data from Fig 3 in ref 78 (Dumitru et al 2019). Going to that paper it is said that the CO2 are modelling results from ref 21 in Dumitru which is a mistake, the correct reference is ref 22 in Dumitru which points to Stap et al. 2016 Earth Planet. Sci. Lett. 439, 1–10 (2016). However, this is just one of many available model-based estimates, which might be right or wrong. Data-based CO2 reconstruction across the M2 event around 3.3 Ma are found in de la Vega et al (2020), <https://doi.org/10.1038/s41598-020-67154-8>. They are quite different and should be taken, if CO2 is plotted here, at minimum in addition to the model-based results.

- I have major problems with the discussion in lines 314-345:

- When referring to Fig 9 be more specific, which subfigure is meant and which time interval, e.g. line 315 "... permafrost disappearance...". When should that be? In Fig 9 we have the cyan colour line at 132.17-132.49 Ma which marks permafrost. Do you mean by its disappearance the end of this line (132.17 Ma)? If so, CO₂ is falling at that point in time, so the opposite of what is discussed. What has δ¹³C of carbonates to do with it? If that means these are data from the ocean implying also a shift in mean ocean δ¹³C, then again it is implied that δ¹³C is getting heavier after permafrost thaw, but δ¹³C in permafrost should be depleted in δ¹³C and its release leads to smaller δ¹³C in the ocean, but also on a time scale shorter than 1 Ma, as explained in my last review that permafrost thaw can impact the C cycle only on short time scales, and not on multi-million years. Furthermore, the whole discussion is just a list of data-based facts which I should see in Fig 9, but as you see I have difficulties in identifying and no content is given. What is the point in showing multi-millennial changes in δ¹³C?

- lines 317 ff: „The global synchronicity (interval 132.5–128 Ma, Fig. 9) of the positive carbon isotope event, the SST variation, and the rise of atmospheric pCO₂ postdating the terrestrial record of plateau permafrost (Fig. 9) collectively points to a strong coupling of the ocean–atmosphere system.“ This sentence again suggests that permafrost thaw has any impact on multi-million years events, which I clearly pointed out in the previous review that this is not possible.

- Similarly, comparing the Cretaceous with the M2 event in the Pliocene is weak, because the data collection (Figs 9-5 to 9-12) is weak or not discussed in detail in the discussion (Figs 9-10, 9-11, 9-12 are never mentioned), CO₂ data in Fig 9.8 is not the best choice. The text (line 325) says ice volume is increasing during M2, but Fig 9-7 shows the opposite (I think the sign is wrong in the Fig 9-7). Actually, if the authors still stick to CO₂ from the model (Stap et al 2016), they should also plot sea level from that paper which goes in the right direction. The author also copy ice volume changes in Fig 9-7 from ref 78. Here, they cherry-pick two out of three given curves in ref 78 without even citing the original sources. Original sources (Rohling et al 2014, Lisiecki & Raymo 2005) should be given here, potentially also showing the 3rd one or given reason why not.

- The wrongness of the comparison cumulates in line 334ff: „In fact, disappearance of Pliocene and Hauterivian permafrost correlated with a global increase of atmospheric pCO₂ and a rise of SST (°C) driving into the mid-Pliocene Warm Period (mPWP), and the early Barremian warm pulse, respectively (Fig. 9).“ In Fig 9-2 permafrost disappears at 132.17 Ma right when CO₂ drops (might change significantly once the CO₂ record is updated), and SST (Fig 9-4) rises about 2 Ma later, thus in complete disconnection to the event.

- The main problem with the comparison of the the Cretaceous and the Pliocene event is the different age scale. One might get into the right direction if both events are plotted on similar time axes and not as done so far comparing 30 Ma year in the Cretaceous with 0.5 Ma in the Pliocene, that is „just“ a factor of 60 difference!!! But I think the best move here would actually be to delete this comparison and Fig 9 altogether. Detailed time series are found in Fig. 7 (which might need some more details in the Figure caption).

REVIEWERS' COMMENTS

Reviewer #1 (Remarks to the Author):

This is my second review of this manuscript. The authors sufficiently addressed the comments and concerns of the reviewers in their response letter. I will only make a few small comments on their rebuttal to my suggestions.

First, I made an error - the reference I suggested they cite was Galloway et al., 2019 (not 2020).

Response to Reviewer:

Done. We changed “2020” to “2019”.

I recommend the authors add the HALIP to Figure 9 as well.

Response to Reviewer:

Following the indications of the Editor we have deleted Fig. 9.

And lastly, please do not cite Galloway et al. 2019 for ice rafted debris - its not a primary reference for this.

Response to Reviewer:

Done.

Grasby et al. 2017 can also be cited for Valanginian glendonite occurrences in the Canadian Arctic.

Response to Reviewer:

Done.

Other than these comments my original review comments remain: the article is well-written, provocative, and should be published in Nature Communications. Sincerely, Jennifer Galloway, Geological Survey of Canada.

Response to Reviewer:

Thanks very much.

Reviewer #2 (Remarks to the Author):

Review on Permafrost in the Cretaceous supergreenhouse

from

Rodríguez-López et al
submitted to Nature Communications
NCOMMS-22-29294A

I am reviewer #2 from the first round of reviews.

The authors followed my main concern of taken out permafrost thaw as an alternative to other long-term carbon cycle processes.

My initial suggestions in the last round of reviews was, that once this main claim is not supported anymore the paper is probably rejected, which is why I did not spend too much time on details.

However, other reviewers and the editorial board seemed to be more positive on the paper at large. Therefore, I try to fill in what I think is still incorrect on the details in the discussion around Figure 9.

- I suggest to replace numbers for addressing the subfigures in Fig 9 by letters, (Fig 9 1-12 -> Fig 9 a-l)

- CO₂ plotted in Fig 9 is not based on the best sources in the literature:

- For the Cretaceous (Fig 9-2) a record was chosen that shows a rapid decline at 132Ma, while the review by Foster et al (2017) DOI: 10.1038/ncomms14845 shows a much more diverse picture. Based on Foster I would argue that you cannot claim any large changes in CO₂ happening at that point in time.

- For the Pliocene (Fig 9-8) the authors plot CO₂ data from Fig 3 in ref 78 (Dumitru et al 2019). Going to that paper it is said that the CO₂ are modelling results from ref 21 in Dumitru which is a mistake, the correct reference is ref 22 in Dumitru which points to Stap et al. 2016 Earth Planet. Sci. Lett. 439, 1–10 (2016). However, this is just one of many available model-based estimates, which might be right or wrong. Data-based CO₂ reconstruction across the M2 event around 3.3 Ma are found in de la Vega et al (2020), <https://doi.org/10.1038/s41598-020-67154-8>. They are quite different and should be taken, if CO₂ is plotted here, at minimum in addition to the model-based results.

- I have major problems with the discussion in lines 314-345:

- When referring to Fig 9 be more specific, which subfigure is meant and which time interval, e.g. line 315 "... permafrost disappearance...". When should that be? In Fig 9 we have the cyan colour line at 132.17-132.49 Ma which marks permafrost. Do you mean by its disappearance the end of this line (132.17 Ma)? If so, CO₂ is falling at that point in time, so the opposite of what is discussed. What has δ¹³C of carbonates to do with it? If that means these are data from the ocean implying also a shift in mean ocean δ¹³C, then again it is implied that δ¹³C is getting heavier after permafrost thaw, but δ¹³C in permafrost should be depleted in δ¹³C and its release leads to smaller δ¹³C in the ocean, but also on a time scale shorter than 1 Ma, as explained in my last review that permafrost thaw can impact the C cycle only on short time scales, and not on multi-million years. Furthermore, the whole discussion is just a list of data-based facts which I should see in Fig 9, but as you see I have difficulties in identifying and no content is given. What is the point in showing multi-millennial changes in δ¹³C?

- lines 317 ff: „The global synchronicity (interval 132.5–128 Ma, Fig. 9) of the positive carbon isotope event, the SST variation, and the rise of atmospheric pCO₂ postdating the terrestrial record of plateau permafrost (Fig. 9) collectively points to a strong coupling of the ocean–atmosphere system.“ This sentence again suggest that

permafrost thaw has any impact on multi-million years events, which I clearly pointed out in the previous review that this is not possible.

- Similarly, comparing the Cretaceous with the M2 event in the Pliocene is weak, because the data collection (Figs 9-5 to 9-12) is weak or not discussed in detail in the discussion (Figs 9-10, 9-11, 9-12 are never mentioned), CO₂ data in Fig 9.8 is not the best choice. The text (line 325) says ice volume is increasing during M2, but Fig 9-7 shows the opposite (I think the sign is wrong in the Fig 9-7). Actually, if the authors still stick to CO₂ from the model (Stap et al 2016), they should also plot sea level from that paper which goes in the right direction. The author also copy ice volume changes in Fig 9-7 from ref 78. Here, they cherry-pick two out of three given curves in ref 78 without even citing the original sources. Original sources (Rohling et al 2014, Lisiecki & Raymo 2005) should be given here, potentially also showing the 3rd one or given reason why not.

- The wrongness of the comparison cumulates in line 334ff: „In fact, disappearance of Pliocene and Hauterivian permafrost correlated with a global increase of atmospheric pCO₂ and a rise of SST (oC) driving into the mid-Pliocene Warm Period (mPWP), and the early Barremian warm pulse, respectively (Fig. 9).“ In Fig 9-2 permafrost disappears at 132.17 Ma right when CO₂ drops (might change significantly once the CO₂ record is updated), and SST (Fig 9-4) rises about 2 Ma later, thus in complete disconnection to the event.

- The main problem with the comparison of the the Cretaceous and the Pliocene event is the different age scale. One might get into the right direction if both events are plotted on similar time axes and not as done so far comparing 30 Ma year in the Cretaceous with 0.5 Ma in the Pliocene, that is „just“ a factor of 60 difference!!! But I think the best move here would actually be to delete this comparison and Fig 9 altogether. Detailed time series are found in Fig. 7 (which might need some more details in the Figure caption).

Response to Reviewer:

Following the indications and recommendations of the Editor, Fig. 9 has been deleted,

and the discussion on the similarities between the Pliocene and the Cretaceous events have been removed from the manuscript.